# Envy-free Policy Teaching to Multiple Agents

**Jiarui Gan**
University of Oxford
jiarui.gan@cs.ox.ac.uk

**Rupak Majumdar**
MPI-SWS
rupak@mpi-sws.org

**Goran Radanovic**
MPI-SWS
gradanovic@mpi-sws.org

**Adish Singla**
MPI-SWS
adish@mpi-sws.org

## Abstract

We study envy-free policy teaching. A number of agents independently explore a common Markov decision process (MDP), but each with their own reward function and discounting rate. A teacher wants to teach a target policy to this diverse group of agents, by means of modifying the agents' reward functions: providing additional bonuses to certain actions, or penalizing them. When personalized reward modification programs are used, an important question is how to design the programs so that the agents think they are treated fairly. We adopt the notion of envy-freeness (EF) from the literature on fair division to formalize this problem and investigate several fundamental questions about the existence of EF solutions in our setting, the computation of cost-minimizing solutions, as well as the price of fairness (PoF), which measures the increase of cost due to the consideration of fairness. We show that 1) an EF solution may not exist if penalties are not allowed in the modifications, but otherwise always exists. 2) Computing a cost-minimizing EF solution can be formulated as convex optimization and hence solved efficiently. 3) The PoF increases but at most quadratically with the geometric sum of the discount factor, and at most linearly with the size of the MDP and the number of agents involved; we present tight asymptotic bounds on the PoF. These results indicate that fairness can be incorporated in multi-agent teaching without significant computational or PoF burdens.

## 1 Introduction

Incentive design is an important approach to influencing rational agents' behavior. In reinforcement learning (RL), the incentive of an agent is expressed through their reward function [1]. One can thus teach a desired policy to an agent by modifying their reward function, in a way that makes the target policy optimal with respect to the modified rewards. In safe RL, for example, penalties can be imposed on dangerous actions to prevent an agent from executing them [2]. In many cases, personalized teaching programs are useful against heterogeneous agents, who might have very different innate reward functions or apply different discounting rate. As a result, the agents may find them rewarded/penalized differently for performing the same action in the same situation (see Figure 1). Concerns of fairness arise, and we ask the question of how to design fair personalized teaching programs so that the agents think that they are treated fairly.

To be more concrete, consider a language teaching setting modeled as an MDP. Each state of the MDP represents the overall skill of a student (agent) and is encoded as the student's performance on different components such as listening, reading, speaking, and writing. Actions available to the students are defined by the levels of effort they put into the components, and it is desired that they always put more effort into the components that they are currently weaker at, which is also the target

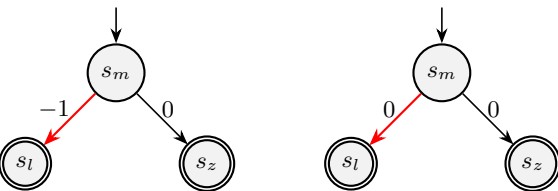

Figure 1: To teach agents to choose the action leading to state $s_l$, an additional reward 1 is necessary for an agent whose innate reward function is the one on the left, whereas an agent with the reward function on the right already finds this target policy optimal, so no additional reward is needed. When these two agents are being taught together, the agent on the right would think they are treated unfairly as they get no bonus for following the target policy while the agent on the left gets bonus 1.

policy the teacher aims to teach. The students' innate reward functions are defined by their interests, which vary across the classroom: some students may be more interested in reading, some enjoy speaking, and some are just not a fan of any of them. The teacher can assign additional credits to incentivize the students to follow the target policy (e.g., credits that can be used to exchange snacks, or that will be considered in the final evaluation). Similar interactions may also happen with other types of training programs in various domains, such as sports training. They can happen both in physical classrooms and virtual classrooms such as language educational apps (e.g., Duolingo uses a credit system where credits can be used to unlock next learning levels). Beyond classroom teaching, examples can also be found in principal-agent settings. For example, a company wants to outsource a task to different contractors. Rewards or penalties are stipulated through customized contracts to ensure that contractors comply with a desired policy when performing the task. Meanwhile, fairness is important as a beneficial factor for long-term partnerships.

## 1.1 Approach and Results

Our first step is to understand what it means to be fair in the setting of policy teaching. Indeed, in a world with growing awareness of equality and transparency, fairness has been discussed and evaluated in a wide range of domains. Various concepts and notions of fairness have been proposed and used [3]. We borrow the well-studied fairness notion of *envy-freeness* (EF) from the literature on fair division. It is a notion that has been used for settling disputes over property divisions or deciding how to split an apartment rent [e.g., 4, 5]. Applying EF to policy teaching, we aim to find a set of personalized teaching programs, such that no agent would prefer to switch the program they receive with another agent. At the same time, as a basic requirement of policy teaching, each program should also incentivize the corresponding agent to use the target policy. Besides the basic version of EF, we also consider two stronger variants: one allows an agent to further deviate from the target policy when evaluating how much they would have got had they been offered another agent's teaching program; the other simply requires all teaching programs to be identical, which is completely fair in a sense.

We investigate several fundamental questions about EF policy teaching.

- *Existence of an EF Solution.* The first question is about the existence of an EF solution under the three EF notions of interest. We show that an EF solution always exists and one can be obtained simply by penalizing undesired actions by a sufficiently large value. Nevertheless, the reverse does not hold true: one cannot hope to find an EF solution only by rewarding actions desired by the target policy. We demonstrate instances that do not admit any EF solution when penalties are not allowed even with the weakest EF notion; we also prove that this non-existence issue is resolved if the agents have the same discount factor.

- *Cost Minimization.* Since reward modification can be very costly, we are also interested in finding out an EF solution with the least cost. We consider the norm of the modification and show that computing a cost-minimizing EF solution can be formulated as convex optimization and can hence be solved efficiently.

- *Price of Fairness.* Finally, we analyze the *price of fairness* (PoF), a quantity that measures the (multiplicative) increase of the cost due to consideration of fairness and is in a similar spirit of the *price of anarchy* (PoA) in game theory [6]. We present tight asymptotic bounds on the PoF. The

PoF increases at most quadratically with the geometric sum of the discount factor and linearly with the size of the MDP in general, while it may also grow linearly with the number of agents involved depending on the specific EF notion considered.

In summary, our results indicate that the consideration of fairness, in addition to the original goal of policy teaching, may result in non-existence of workable solutions but the existence is guaranteed in a fairly wide range of important settings. It does not appear to increase the computational complexity of policy teaching, while the additional cost it incurs grows moderately with the size of the problem. The results indicate that fairness can be incorporated in multi-agent teaching without significant computational or PoF burdens.

## 1.2 Related Work

Our work lies at the intersection of policy teaching and envy-free resource allocation.

**Policy Teaching** Without the fairness constraints, our model can be seen as a policy teaching problem for each individual agent in the model. A number of studies have looked at this problem [7, 8]. The problem can be computationally harder though when the target is to hit one in a set of policies rather than a single target [9]. When the teacher is targeting a malicious policy, policy teaching can also be interpreted as reward poisoning [10, 11, 12, 13, 14]. From a technical point of view, these two problems are almost identical and can be solved by using the same techniques. However, conceptually, it is less likely that one would take fairness into consideration when designing a poisoning attack. More broadly, policy teaching can be seen as a sub-field of reward design, a broader area that studies how to influence agents' behaviors thought tweaking the reward function. The objectives of these studies are not limited to inducing a target policy. A notable example is reward shaping [15, 16, 17], which aims to accelerate an agent's learning process through reward design. Indeed, while our focus is on policy teaching, the same question of how to design rewards fairly can be asked with other objectives as well. These can be potential directions for future work.

**Fair Division** The study of fair division dates back to the early work of Foley [18], and the formal concept of envy-freeness appeared even earlier [19]. Research on fair division has since evolved into a large body of work, with focuses on allocation of divisible or indivisible items [20, 21, 22, 23]. Our work is in particular related to fair allocation of indivisible goods with subsidies [24], where external benefits are provided to change the agents' original incentives. The difference is that no items are allocated in our model and our goal in addition to achieving fairness is to teach the target policy.

We note that there are also other studies on machine teaching settings involving multiple agents or multiple teachers [25, 26, 27], though with very different models from ours. From a mechanism design perspective, our model can also be viewed as one version of the contract design problem [28], where a principal offers an agent a contract for performing a target policy, but might be uncertain about the agent's type (i.e., the original reward function). Our EF solutions correspond exactly to truthful mechanisms that elicit the agent's true type.

## 2 Preliminaries

There are $n$ agents $1, \ldots, n$. Let $[n] = \{1, \ldots, n\}$. Each agent $i \in [n]$ faces an MDP $\mathcal{M}_i = \langle S, A, R_i, P, \mathbf{z}, \gamma_i \rangle$. The MDPs have the same state space $S$, action space $A$, transition function $P : S \times A \times S \to [0, 1]$, and initial state distribution $\mathbf{z}$. Moreover, there is a reward function $R_i : S \times A \to \mathbb{R}$ and discount factor $\gamma_i$ for each agent $i \in [n]$. Whenever agent $i$ takes an action $a$ in state $s$, a reward $R_i(s, a)$ is generated for this agent; meanwhile the state transitions to $s' \in S$ with probability $P(s, a, s')$. We consider the setting where each agent is concerned with the (expected) cumulative reward, i.e., the discounted sum of rewards with respect to the factor $\gamma_i$, obtained over an infinite horizon. More specifically, the cumulative reward of agent $i$ for executing a policy $\pi : S \to \Delta(A)$ is

$$\rho_i^\pi = \mathbb{E}\left[ \sum_{t=0}^{\infty} (\gamma_i)^t \cdot R_i(s_t, a_t) \,\middle|\, s_0 \sim \mathbf{z}, \pi \right],$$

where the expectation is taken over the trajectory $(s_t, a_t)_{t=0}^{\infty}$ resulting from an initial state $s_0$ sampled from $\mathbf{z}$ and the agent executing $\pi$ subsequently. Each agent aims to find an optimal policy, which

maximizes $\rho_i^\pi$, and this can usually be handled by standard planning and reinforcement learning algorithms.

Throughout the paper, we consider the setting where the agents operate independently in separate environments. Their payoffs are only determined by their own policies.

## 2.1 Single-agent Policy Teaching

Consider the situation where we want an agent $i$ to execute a target policy $\pi^\star$, but the agent finds a different policy $\pi'$ optimal for $\mathcal{M}_i$. To incentivize the agent to use $\pi^\star$, a typical way is to modify the the reward function by providing additional rewards (positive or negative). We follow the literature and consider only deterministic target policies. (Indeed, in general, one cannot hope to incentivize an agent to use a non-deterministic policy only by tweaking the reward function.)

Specifically, the teacher chooses a *reward adjustment function* $\delta_i : S \times A \to \mathbb{R}$, or *adjustment* for short, whereby an additional reward $\delta_i(s, a)$ is provided whenever the agent takes an action $a \in A$ in a state $s \in S$. Effectively, the adjustment changes the agent's reward function to $\widetilde{R}_i(s, a) = R_i(s, a) + \delta_i(s, a)$. The agent then optimizes their policy with respect to $\widetilde{R}_i$, and will be incentivized to use $\pi^\star$ if it offers the maximum payoff (cumulative reward) with respect to $\widetilde{R}_i$. We can view each agent's payoff for policy $\pi$ as a function of $\delta_i$ as follows:

$$\rho_i^\pi(\delta_i) := \mathbb{E}\left[ \sum_{t=0}^\infty (\gamma_i)^t \cdot \widetilde{R}_i(s_t, a_t) \,\middle|\, s_0 \sim \mathbf{z}, \pi \right].$$

Moreover, we define the V-function and Q-function of $\pi$ given adjustment $\delta_i$ as:

$$V_i^\pi(s \mid \delta_i) = Q_i^\pi(s, \pi(s) \mid \delta_i),$$
$$\text{and} \quad Q_i^\pi(s, a \mid \delta_i) = \widetilde{R}_i(s, a) + \gamma_i \cdot \mathbb{E}_{s' \sim P(s, a, \cdot)} V_i^\pi(s' \mid \delta_i).$$

The V-function captures the expected cumulative reward by starting from $s$ and following $\pi$. The Q-function captures the expected cumulative reward by starting from $s$, taking action $a$ at the first step, and following $\pi$ subsequently. We have

$$\rho_i^\pi(\delta_i) = V_i^\pi(\mathbf{z} \mid \delta_i) := \mathbb{E}_{s_0 \sim \mathbf{z}} V_i^\pi(s_0 \mid \delta_i).$$

Using these two functions, the Bellman equation further characterizes the optimal policy in the MDP: a policy $\pi$ is optimal if and only if the following Bellman optimality equation holds: $Q_i^\pi(s, \pi(s) \mid \delta_i) \geq Q_i^\pi(s, a \mid \delta_i)$ for all $s \in S$ and $a \in A$.

**Incentive Constraints** Hence, the goal of policy teaching is to make the target policy $\pi^\star$ a solution to the Bellman optimality equation. Since the agent may find multiple policies optimal, a robustness guarantee $\epsilon > 0$ is imposed to *strictly* incentivize the agent to use $\pi^\star$, and this results in the following incentive constraints:

$$Q_i^{\pi^\star}(s, \pi^\star(s) \mid \delta_i) \geq Q_i^{\pi^\star}(s, a \mid \delta_i) + \epsilon \quad \text{for all } a \neq \pi^\star(s). \tag{1}$$

The constraints ensure the optimality of $\pi^\star$ even if there is a small error in the Q-values.

**Cost Measures** In addition to incentivizing $\pi^\star$, the teacher also wants to find the most cost-efficient way of teaching. We consider the norm of the adjustment, which means the following cost measure:

$$\text{cost}(\delta_i) = \|\delta_i\| := \left( \sum_{s \in S, a \in A} (\delta_i(s, a))^2 \right)^{1/2}. \tag{2}$$

## 3 Teaching Multiple Agents and EFness

In the multi-agent setting, the teacher provides an adjustment to every agent in $[n]$. We call a collection of adjustments $(\delta_i)_{i \in [n]}$ an *adjustment scheme*. A basic approach for this setting is to deal with each agent separately, by solving a single-agent teaching problem for each agent. The solution obtained via this approach provides personalized adjustments to the agents and it minimizes the

teacher's total cost. Nevertheless, it might not be fair as we showed in the example of Figure 1. To be more specific, we define three fairness notions, each being stronger than the previous one. We start with the following weak EF notion.

**Definition 3.1** (**Weak envy-freeness (WEF)**). An adjustment scheme $(\delta_i)_{i \in [n]}$ is *weakly envy-free* if it holds for all $i \in [n]$ that

$$\rho_i^{\pi^\star}(\delta_i) \geq \rho_i^{\pi^\star}(\delta_j) \qquad \text{for all } j \in [n]. \tag{3}$$

In other words, no agent $i$ would prefer the adjustment for another agent $j$ to their own.

The above notion only compares the agents' benefits under $\pi^\star$. When $\delta_i$ incentivizes agent $i$ to use $\pi^\star$, the left side of (3) is also exactly the highest possible benefit $i$ can obtain given adjustment $\delta_i$. But this is not true for the adjustment on the right side: $\pi^\star$ need not be optimal for agent $i$ with respect to $R_i + \delta_j$; a higher cumulative reward might be attainable if the agent switches to another policy. In some scenarios, this higher potential reward may be a legitimate concern when fairness is evaluated. The following stronger notion takes this aspect into account.

**Definition 3.2** (**Envy-freeness (EF)**). An adjustment scheme $(\delta_i)_{i \in [n]}$ is *envy-free* if it holds for all $i \in [n]$ that:

$$\rho_i^{\pi^\star}(\delta_i) \geq \max_{\pi} \rho_i^{\pi}(\delta_j) \quad \text{ for all } j \in [n]. \tag{4}$$

An even stronger fairness notion defined below simply requires the same adjustment to be applied to all the agents. It is completely fair in a sense.

**Definition 3.3** (**Strong envy-freeness (SEF)**). An adjustment scheme $(\delta_i)_{i \in [n]}$ is *strongly envy-free* if $\delta_i = \delta_j$ for all $i, j \in [n]$.

Let $\mathcal{D}_{\text{WEF}}$, $\mathcal{D}_{\text{EF}}$, and $\mathcal{D}_{\text{SEF}}$ denote the sets of adjustment schemes complying with the above fairness notions respectively. It is not hard to see that: $\mathcal{D}_{\text{WEF}} \supseteq \mathcal{D}_{\text{EF}} \supseteq \mathcal{D}_{\text{SEF}}$.

Besides achieving EFness, the original goal of policy teaching is to incentivize the agents to use $\pi^\star$. Hence, we will call adjustment schemes that satisfy equation (1) *feasible* schemes (Definition 3.4). Indeed, the definitions of WEF and SEF would be meaningless without the feasibility requirement, in which case they can be achieved trivially by providing zero additional reward to every agent. (The definition of EF, on the other hand, already incorporates the incentive constraints as equation (3) also includes the case where $i = j$, except that there is no $\epsilon$ robustness requirement.)

**Definition 3.4** (**Feasibility**). An adjustment scheme $(\delta_i)_{i \in [n]}$ is *feasible* (with respect to a robustness guarantee $\epsilon > 0$) if equation (1) holds for all $i \in [n]$.

Sometimes only bonuses (non-negative additional rewards) are allowed, e.g., when one can provide the agents with subsidies but cannot penalize them. Hence, we are also interested in finding *non-negative* adjustment schemes defined as follows.

**Definition 3.5** (**Non-negativity**). An adjustment scheme $(\delta_i)_{i \in [n]}$ is *non-negative* if $\delta_i(s, a) \geq 0$ for all $i \in [n]$, $s \in S$, and $a \in A$.

Similarly to the single-agent policy teaching problem, cost-minimizing solutions are desired. We consider the sum of the teaching costs in the multi-agent setting:

$$\text{cost}(\delta) = \sum_{i \in [n]} \text{cost}(\delta_i).$$

## 4 Existence of Fair Solutions

Before we delve into the computation of a cost-minimizing solution, we first investigate the existence of a solution with respect to the above defined fairness notions and requirements. Throughout this section, we assume that the original rewards are bounded in the interval $[-h, h]$, i.e., $R_i(s, a) \in [-h, h]$ for all $s$, $a$, and $i$. Our first result shows that a fair and feasible solution always exists under all of the above fairness notions, in particular under the strongest notion SEF.

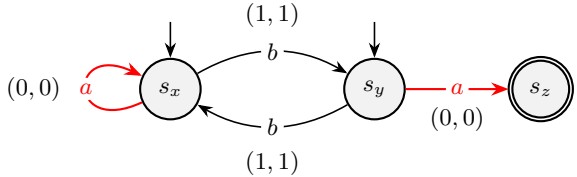

Figure 2: There are two agents, whose discount factors are $\gamma_1 = 0.9$ and $\gamma_2 = 0.5$, respectively. $S = \{s_x, s_y, s_z\}$, $A = \{a, b\}$, and all transitions are deterministic. Originally, the agents' reward functions are the same and their rewards are annotated as vectors on the edges, with $R_1(s, a) = R_2(s, a) = 0$ and $R_1(s, b) = R_2(s, b) = 1$ for all $s \in S$. The states $s_x$ and $s_y$ are chosen as the initial state with equal probability. The target policy $\pi^\star$, highlighted in red, is such that $\pi^\star(s) = a$ for all $s \in S$ (i.e., it always selects action $a$).

**Theorem 4.1.** *For any robustness guarantee $\epsilon > 0$, an SEF and feasible adjustment scheme always exists.*[1]

*Proof sketch.* The idea is to penalize actions not following the target policy by a sufficiently large value. We construct an adjustment scheme $(\delta_i)_{i \in i}$ where

$$\delta_i(s, a) = \begin{cases} 0, & \text{if } a = \pi^\star(s) \\ -\max_{i' \in [n]} \frac{2h}{1 - \gamma_{i'}} - \epsilon, & \text{otherwise} \end{cases}$$

for all $s \in S$ and $i \in [n]$. The scheme is obviously SEF as $\delta_i$ is the same for all the agents. It can also be verified that it is feasible. Intuitively, the penalty is so large such that once the agent is penalized, the subsequent cumulative rewards cannot compensate for the loss due to this penalty even if the highest rewards are attained at every subsequent step. □

Nevertheless, the reverse is not true. If we only allow non-negative schemes, the existence of a feasible solution cannot be taken for granted, and in general one cannot hope to teach a target policy by placing large bonuses on actions following the target policy. As we prove in Theorem 4.2, the example illustrated in Figure 2 does not admit any EF feasible solution (and hence neither an SEF one), even though it involves only two agents and the agents have the same reward function (but different discount factors).

**Theorem 4.2.** *For any robust guarantee $\epsilon \geq 0$, a feasible adjustment scheme that is WEF and non-negative may not exist, even when there are only two agents and their reward functions are the same.*

*Proof.* We show that there exists no feasible adjustment scheme that is WEF and non-negative in the example illustrated in Figure 2. Suppose for the sake of contradiction that there exists a scheme $(\delta_1, \delta_2)$ which is EF, non-negative, and feasible.

Without loss of generality, we can assume that $\delta_1(s, b) = \delta_2(s, b) = 0$ for all $s \in S$. Indeed, it is not hard to see that if there exists a WEF and feasible scheme with some or all of these values being strictly positive, it will remain WEF and feasible if these values are reset to 0. Hence, it remains to pin down the values for action $a$ in the adjustment scheme. For ease of description, let $x_i = \delta_i(s_x, a)$ and $y_i = \delta_i(s_y, a)$ for $i \in \{1, 2\}$.

We first argue that the following two inequalities hold:

$$x_1 \geq x_2, \quad \text{and} \quad y_2 \geq y_1. \tag{5}$$

To see this, consider the WEF constraints defined in (3). The adjustment scheme considered is WEF, so $\rho_i^{\pi^\star}(\delta_i) \geq \rho_i^{\pi^\star}(\delta_{-i})$, where $-i$ is the index in $\{1, 2\}$ that is different from $i$. Hence,

$$0.5 \cdot V_i^{\pi^\star}(s_x \mid \delta_i) + 0.5 \cdot V_i^{\pi^\star}(s_y \mid \delta_i) \geq 0.5 \cdot V_i^{\pi^\star}(s_x \mid \delta_{-i}) + 0.5 \cdot V_i^{\pi^\star}(s_y \mid \delta_{-i}), \tag{6}$$

---

[1]Full proofs and omitted proofs can all be found in the appendix.

where $0.5$ is the probability in the initial distribution. It is easy to derive the V-values of $s_x$ and $s_y$ under $\pi^\star$ as neither of them depends on the V-values of any other states. We have

$$V_i^{\pi^\star}(s_x \mid \delta_j) = Q_i^{\pi^\star}(s_x, a \mid \delta_j) = \tfrac{1}{1-\gamma_i} \cdot x_j, \tag{7}$$

$$\text{and} \quad V_i^{\pi^\star}(s_y \mid \delta_j) = Q_i^{\pi^\star}(s_y, a \mid \delta_j) = y_j. \tag{8}$$

Plugging these two equations back into (6) gives

$$0.5 \cdot \tfrac{1}{1-\gamma_i} \cdot x_i + 0.5 \cdot y_i \geq 0.5 \cdot \tfrac{1}{1-\gamma_{-i}} \cdot x_{-i} + 0.5 \cdot y_{-i}.$$

Replacing $\gamma_i$ with the corresponding values gives

$$10 \cdot x_1 + y_1 \geq 10 \cdot x_2 + y_2 \tag{9}$$
$$2 \cdot x_2 + y_2 \geq \phantom{0} 2 \cdot x_1 + y_1 \tag{10}$$

Hence, (9)+(10) gives $x_1 \geq x_2$, and (9)+5×(10) gives $y_2 \geq y_1$.

Next, we turn to the feasibility constraints. The assumption that $\delta_i$ is feasible means that

$$Q_i^{\pi^\star}(s_x, a \mid \delta_i) \geq Q_i^{\pi^\star}(s_x, b \mid \delta_i) + \epsilon = 1 + \gamma_i \cdot V_i^{\pi^\star}(s_y \mid \delta_i) + \epsilon$$

$$\text{and} \quad Q_i^{\pi^\star}(s_y, a \mid \delta_i) \geq Q_i^{\pi^\star}(s_y, b \mid \delta_i) + \epsilon = 1 + \gamma_i \cdot V_i^{\pi^\star}(s_x \mid \delta_i) + \epsilon$$

Substituting (7) and (8) into the above two equations gives:

$$1 + \tfrac{\gamma_i}{1-\gamma_i} \cdot x_i < y_i < \tfrac{1}{\gamma_i(1-\gamma_i)} \cdot x_i - \tfrac{1}{\gamma_i}. \tag{11}$$

Using (5) and (11), we get that

$$9 \cdot x_1 + 1 < y_1 \leq y_2 < 4 \cdot x_2 - 2 \leq 4 \cdot x_1 - 2.$$

This means that $x_1 < 0$ and contradicts the assumption that $\delta$ is a non-negative scheme. $\square$

It turns out that the agents' discount factors play a crucial role: an identical discount factor is sufficient for ensuring the existence of a feasible SEF solution. We present this result below.

**Theorem 4.3.** *When the agents have the same discount factor, a feasible adjustment scheme that is also SEF and non-negative always exists, for any robustness guarantee $\epsilon > 0$.*

*Proof sketch.* Suppose that $\gamma_1 = \cdots = \gamma_n = \gamma$. Let $H = \tfrac{2}{1-\gamma} \cdot h + \epsilon$. We construct the following scheme $\delta = (\delta_i)_{i \in [n]}$:

$$\delta_i(s, a) = \begin{cases} H + \tfrac{\gamma}{1-\gamma} \cdot H \cdot \sum_{s' \in S^{\mathrm{T}}} P(s, a, s'), & \text{if } a = \pi^\star(s) \\ 0, & \text{otherwise} \end{cases}$$

for all $s \in S$ and $i \in [n]$, where $S^{\mathrm{T}}$ denotes the set of terminal states in $S$.

The scheme is obviously non-negative and SEF, so it remains to argue that it is feasible. Intuitively, $\delta_i$ results in the agent receiving a reward that is sufficiently large (and is roughly the same) at every step if the agent follows $\pi^\star$. Rewards are adjusted by a factor of $1/(1-\gamma)$ at the subsequent terminal states so that it is as if the process continues forever with the same reward $H$ generated at every step (but the cumulative reward $\tfrac{1}{1-\gamma} \cdot H$ is paid off at once). Therefore, under $\delta_i$, the process is equivalent to an infinite-horizon process where the agent gets a (roughly) constant positive reward $H$ at every step if the agent follows $\pi^\star$. This loss due to not following $\pi^\star$ at some step is $H$, and it is sufficiently large so that the optimal choice for the agent in such a process is to always follow $\pi^\star$. $\square$

## 5 Computing an Optimal Fair Solution

In terms of the computation of a cost-minimization fair solution, our main result is as follows. For each of the EF notions we defined above, the set of fair solutions lie in a convex polytope defined by polynomially many linear constraints. Hence, to find out a cost minimizing solution can be formulated as a convex optimization problem given that the cost function (2) is a convex function of the adjustment scheme. We show how the various types of constraints that need to be incorporated can be written as linear constraints next.

**Feasibility Constraints** A feasible scheme is characterized by the following linear constraints, where in addition to the variables $\delta_i(s,a)$ encoding the adjustment scheme, we add an auxiliary variable $V_i(s)$ for each $s \in S$, and $Q_i(s,a)$ for each pair $(s,a) \in S \times A$. The auxiliary variables correspond to the V- and Q-functions of the target policy when $\delta$ is applied.

$$V_i(s) = Q_i(s, \pi^\star(s)) \qquad\qquad\qquad \text{for all } i, s \qquad (12a)$$

$$Q_i(s,a) = R_i(s,a) + \delta_i(s,a) + \gamma_i \sum_{s' \in S} P(s,a,s') \cdot V_i(s') \qquad \text{for all } i, s, a \qquad (12b)$$

$$Q_i(s, \pi^\star(s)) \geq Q_i(s,a) + \epsilon \qquad\qquad \text{for all } i, s, a \neq \pi^\star(s) \qquad (12c)$$

Specifically, the first two lines follow from the Bellman equation and capture the values $V_i^{\pi^\star}(s \mid \delta_i)$ and $Q_i^{\pi^\star}(s, a \mid \delta_i)$; the last line is the incentive constraints and enforces $\delta$ to be feasible.

Next, we consider each of the fairness notions.

**SEF Constraints** To enforce SEF simply amounts to the following constraints for each pair of agents $i, j \in [n]$, which enforces the schemes to be identical.

$$\delta_i(s,a) = \delta_j(s,a) \qquad\qquad\qquad \text{for all } s, a \qquad (13)$$

**WEF Constraints** To enforce WEF, we add variables $V_{i,j}$ and $Q_{i,j}$ to capture the values $V_i^{\pi^\star}(s \mid \delta_j)$ and $Q_i^{\pi^\star}(s, a \mid \delta_j)$, i.e., the values agent $i$ would have got had they been offered the adjustment for agent $j$. Then we add the following constraints, which are similar to the Bellman equation, so that these additional variables acquire the desired values.

$$V_{i,j}(s) = Q_{i,j}(s, \pi^\star(s)) \qquad\qquad\qquad \text{for all } i, j, s \qquad (14a)$$

$$Q_{i,j}(s,a) = R_i(s,a) + \delta_j(s,a) + \gamma_i \sum_{s' \in S} P(s,a,s') \cdot V_{i,j}(s') \qquad \text{for all } i, j, s, a \qquad (14b)$$

Thus, WEF simply amounts to the following constraints for each pair of agents $i, j \in [n]$ (recall that $\mathbf{z}$ is the distribution of the initial state):

$$\sum_{s \in S} z_s \cdot V_i(s) \geq \sum_{s \in S} z_s \cdot V_{i,j}(s) \qquad\qquad \text{for all } s, a \qquad (15)$$

**EF Constraints** Similarly to the approach for handling the WEF constraints, we need additional variables to capture the values of each agent $i$ had they been offered adjustment $\delta_j$. Indeed, we also use the constraints in (14) and (15) but replace (14a) with the following one:

$$V_{i,j}(s) \geq Q_{i,j}(s,a) \qquad\qquad\qquad \text{for all } i, j, s, a \qquad (16a)$$

which associates $V_{i,j}(s)$ to the maximum $Q_{i,j}(s,a)$, instead of $Q_{i,j}(s, \pi^\star(s))$. Note that under these constraints, the value of $V_{i,j}(s)$ in a solution is not necessarily equal to the V-value of $s$ under the optimal policy; it is only an upper bound of them. This will not cause any issue to the approach since the solution is EF if and only if (15) holds for some upper bounds $V_{i,j}(s)$ of $V_i^{\pi^\star}(s \mid \delta_j)$.

**Non-negativity Constraints** Finally, to enforce non-negativity, we simply need the additional constraint: $\delta_i(s,a) \geq 0$ for all $i$, $s$, and $a$.

# 6 Price of Fairness

We now consider the price of fairness (PoF). The PoF measures the increase of teaching cost due to consideration of fairness. In a similar spirit to the celebrated concept of the price of anarchy (PoA) in game theory, the PoF compares the ratio between the minimum costs with and without fairness constraints. We define the PoWEF, PoEF, and PoSEF for our three fairness notions, which stand for the prices of WEF, EF, and SEF, respectively. Formally, let $\mathcal{I}_{n,m,\lambda}$ be the set of instances with $n$ agents, $m$ state-action pairs (i.e., $m = |S| \cdot |A|$), and $\frac{1}{1-\gamma_i} \leq \lambda$ for all $i \in [n]$. We define

$$\text{PoEF}(n, m, \lambda) := \max_{I \in \mathcal{I}_{n,m,\lambda}} \frac{\min_{\delta:\ \text{EF and feasible for } I} \text{cost}(\delta)}{\min_{\delta:\ \text{feasible for } I} \text{cost}(\delta)}.$$

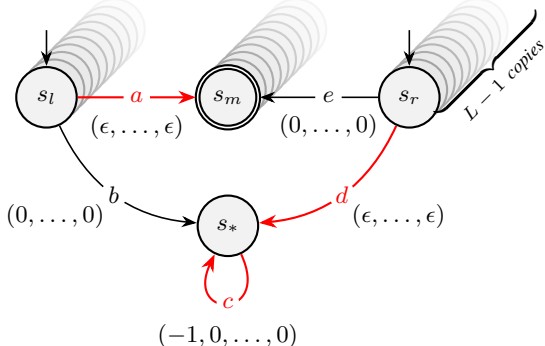

Figure 3: There are $n$ agents with discount factors $\gamma_1 = \cdots = \gamma_n = \gamma$. $A = \{a, b, c, d, e\}$ and all transitions are deterministic. The initial rewards are annotated at the corresponding edges, and they are the same for agents $2, \ldots, n$ (agent 1 has a different reward for action $c$). There are $L - 1$ sets of additional copies of $s_l$, $s_m$, and $s_r$. Every copy of $s_l$ and $s_r$ is connected to the copy of $s_m$ in the same set. In addition, copies of $s_l$ and $s_r$ are also connected to $s_*$ (who has no copies). Each new connection has the same initial rewards as its original copy. The initial state follows a uniform distribution over $s_l$, $s_m$, and all their copies. The target policy is highlighted in red: $\pi^\star(s_l) = a$, $\pi^\star(s_r) = d$, and $\pi^\star(s_*) = c$ (and the same for the corresponding copies).

Namely, the value indicates how large the price can be for instances at the same scale. The PoWEF and PoSEF can be defined in the same way with the corresponding notions.

We analyze the asymptotic growth of the PoF as functions of $n$, $m$, and $\lambda$. The results are presented in Theorem 6.1 and all the bounds are tight. The PoF increases linearly with $\lambda$ and sublinearly with the size of the MDP in all the cases, and the PoEF and PoSEF also grows linearly with the number of agents involved.

**Theorem 6.1.** $\text{PoWEF}(n, m, \lambda) = \Theta(\lambda \cdot \sqrt{m})$, $\text{PoEF}(n, m, \lambda) = \Theta(\lambda \cdot n \cdot \sqrt{m})$, and $\text{PoSEF}(n, m, \lambda) = \Theta(\lambda \cdot n \cdot \sqrt{m})$.

Due to space limit, we leave the detailed proofs of the PoF bounds to the appendix and only provide some intuition about the bounds here. The lower bounds are obtained with the hard instances illustrated in Figure 3. Without fairness consideration, all agents except agent 1 already find the target policy optimal, whereas agent 1 prefers action $e$ to $d$ at state $s_r$. Hence, it suffices to give agent 1 a bonus of 1 for taking action $c$, and the overall cost is 1. Now consider the fairness constraints and suppose that we still provide a bonus $\delta_1(s_*, c) = 1$. The consequence is that agents $2, \ldots, n$ will be envious of this bonus to agent 1. To achieve SEF for example, the same bonus will have to be offered to these agents as well. However, a bonus on $c$ will also incentivize the agents to take action $b$ instead of $a$, leading to violation of the feasibility constraint. Inevitably, to construct a feasible and fair in this example, we cannot hope to only modify the reward for action $c$ (and only the reward for agent 1 when EF and SEF are considered). Modifying the other rewards is however much more costly since each one of them has $L - 1$ copies of themselves, which requires the same modification by symmetry.

To derive the upper bounds, for the PoWEF we construct the following adjustment scheme $\delta = (\delta_i)_{i \in [n]}$ in a similar approach to proving the existence of a fair solution in Theorem 4.1:

$$\delta_i(s, x) = \begin{cases} 0, & \text{if } x = \pi^\star(s) \\ -\frac{2}{1-\gamma_i} \cdot C_i, & \text{otherwise} \end{cases}$$

where $C_i$ denotes the minimum cost for teaching agent $i$ when fairness is not considered. Let $\widehat{\delta}_i$ be the adjustment achieving the minimum cost for each $i$. It can be easily verified that $\frac{\|\delta_i\|}{\|\widehat{\delta}_i\|} \leq 2\lambda \cdot \sqrt{m}$ for all $i \in [n]$, and hence $\text{PoWEF}(n, m, \lambda) \leq \frac{\sum_{i \in [n]} \|\delta_i\|}{\sum_{i \in [n]} \|\widehat{\delta}_i\|} = O(\lambda \cdot \sqrt{m})$. Moreover, $\delta$ is WEF since the scheme only penalizes actions that do not follow the target policy. The argument for showing that $\delta$ is feasible is more involved and we leave it to the appendix. A similar approach can be used to derive the upper bounds of the PoEF and the PoSEF, where we penalize actions that do not follow the policy even more, by $-\max_{j \in [n]} \frac{3}{1-\gamma_j} \cdot C_j$. This also leads to a dependency on $n$ in the bound.

|  | PoWEF | PoEF | PoSEF |
|---|---|---|---|
| No restriction | $\Theta(\lambda \cdot \sqrt{m})$ | $\Theta(\lambda \cdot n \cdot \sqrt{m})$ | $\Theta(\lambda \cdot n \cdot \sqrt{m})$ |
| Non-neg & identical $\gamma$ | $\Theta(\lambda \cdot n \cdot \sqrt{m})$ | $\Theta(\lambda^2 \cdot n \cdot \sqrt{m})$ | $\Theta(\lambda^2 \cdot n \cdot \sqrt{m})$ |

Table 1: Summary of the PoF.

## 6.1 PoF with Non-negative Adjustments

We also investigate PoF with non-negative adjustments and compare the costs of the best non-negative adjustment schemes with and without the fairness constraints. Since a feasible and fair solution may not exist with non-negative adjustments, we analyze the case where the agents have the same discount factor. The existence of a feasible fair solution is guaranteed in this case according to Theorem 4.3. The PoF bounds are presented in Theorem 6.2, where the PoWEF now also depend on the number of agents, and the bounds of the PoEF and PoSEF depend quadratically on $\lambda$.

**Theorem 6.2.** *When the scheme is required to be non-negative and all the agents have the same discount factor, it holds that* $\mathrm{PoWEF}(n, m, \lambda) = \Theta(\lambda \cdot n \cdot \sqrt{m})$, $\mathrm{PoEF}(n, m, \lambda) = \Theta(\lambda^2 \cdot n \cdot \sqrt{m})$, *and* $\mathrm{PoSEF}(n, m, \lambda) = \Theta(\lambda^2 \cdot n \cdot \sqrt{m})$.

The reason why the lower bound of the PoWEF now increases with $n$ can be seen intuitively from the instances in Figure 3, too. Now that the adjustments must be non-negative, to incentivize agent 1 to choose action $c$, a bonus of at least 1 has to be offered. Accordingly, in order for agent 2 (or any agent $i \geq 2$) to not envy this bonus, additional bonuses must be offered to them as well, resulting in a growth with the number of agents. (Without non-negativity, we can penalize agent 1 instead of offering agents $2, \ldots, n$ bonuses to avoid a dependency on $n$ in the lower bound of the PoWEF.) The proofs of the bounds can be found in the appendix and all the PoF bounds are summarized in Table 1.

## 7 Conclusion

We studied the fairness issue in policy teaching and adopted the notion of envy-freeness to formalize the problem. Several fundamental questions regarding the existence of a fair solution, the computation of cost-minimization solution, and the price of considering fairness have been answered in the paper. For future work, it would be interesting to generalize the model to other reward design settings, where a larger set of design objectives or cost measures can be considered. For example, one can use the cumulative payment of the teacher as the cost measure. Indeed, since the cumulative payment is a linear function of the adjustments, the same computation approach we presented applies by replacing the objective function, whereby we obtain a linear program. In terms of the PoF bounds, in a previous version of this work we conjectured that similar bounds can be derived with the cumulative payment cost measure, but it turns out the PoF might also depend on other factors such as the initial state distribution. A detailed analysis of the bounds is an interesting direction for future work.

**Limitations** As we mentioned earlier in the paper, policy teaching is equivalent to reward poisoning from a technical point of view. Hence, almost any techniques that applies to policy teaching also applies immediately to solve reward poisoning problems. We note this potential negative social impact of our results but also remark that since our consideration is fairness we are not aware of any scenario where a malicious party considers fairness when launching a poisoning attack. There are many other notions of fairness, equity, and equality. The EF notions we studied are concerned with the additional rewards provided by the adjustment scheme but not with the overall rewards. Hence, they are not applicable if the latter should be the key consideration.

## Acknowledgments and Disclosure of Funding

The authors thank Warut Suksompong for useful suggestions on the related work and the anonymous reviewers for their insightful comments.

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
