# A   Existence of Fair Solutions

**Theorem 4.1.** *For any robustness guarantee $\epsilon > 0$, an SEF and feasible adjustment scheme always exists.*[2]

*Proof.* The idea is to penalize actions off the target policy by a sufficiently large value. We construct an adjustment scheme $(\delta_i)_{i \in i}$ where

$$\delta_i(s, a) = \begin{cases} 0, & \text{if } a = \pi^\star(s) \\ -\max_{i' \in [n]} \frac{2h}{1-\gamma_{i'}} - \epsilon, & \text{otherwise} \end{cases}$$

for all $s \in S$ and $i \in [n]$. The scheme is SEF as $\delta_i$ is the same for all the agents.

To see that it is also feasible, observe that by following the target policy $\pi^\star$, an agent obtains reward at least $-h$ in every step. Hence, for all $s \in S$ and all $a \neq \pi^\star(s)$, we have

$$Q_i^{\pi^\star}(s, \pi^\star(s) \mid \delta_i) \geq -\frac{h}{1-\gamma_i} \geq -\max_{i' \in [n]} \frac{h}{1-\gamma_{i'}}.$$

It then follows that

$$Q_i^{\pi^\star}(s, \pi^\star(s) \mid \delta_i) \geq \delta_i(s, a) + \frac{h}{1-\gamma_i} + \epsilon$$

$$\geq \delta_i(s, a) + \gamma_i \cdot \sum_{s' \in S} P(s, a, s') \cdot V_i^{\pi^\star}(s' \mid \delta_i) + \epsilon$$

$$= Q_i^{\pi^\star}(s, a \mid \delta_i) + \epsilon,$$

where we used the fact that $V_i^{\pi^\star}(s' \mid \delta_i) \leq \frac{h}{1-\gamma_i}$ for all $s'$, which is due to the fact that the reward obtained at every step is at most $h$. □

**Theorem 4.3.** *When the agents have the same discount factor, a feasible adjustment scheme that is also SEF and non-negative always exists, for any robustness guarantee $\epsilon > 0$.*

*Proof.* Suppose that $\gamma_1 = \cdots = \gamma_n = \gamma$. Let $H = \frac{2}{1-\gamma} \cdot h + \epsilon$. We construct the following scheme $\delta = (\delta_i)_{i \in [n]}$:

$$\delta_i(s, a) = \begin{cases} H + \frac{\gamma}{1-\gamma} \cdot H \cdot \sum_{s' \in S^\mathrm{T}} P(s, a, s'), & \text{if } a = \pi^\star(s) \\ 0, & \text{otherwise} \end{cases} \tag{17}$$

for all $s \in S$ and $i \in [n]$, where $S^\mathrm{T}$ denotes the set of terminal states in $S$. The scheme is obviously non-negative and SEF. We show that it is also feasible.

Consider an arbitrary agent $i$. We first argue that $V_i^{\pi^\star}(s \mid \delta_i) \in \left[\frac{H-h}{1-\gamma}, \frac{H+h}{1-\gamma}\right]$ for all $s \in S \setminus S^\mathrm{T}$. Indeed, if the original reward function $R_i$ was a zero function ($R_i(s, a) = 0$), it can be easily verified that the solution to the Bellman equation would be: $V_i^{\pi^\star}(s \mid \delta_i) = \frac{H}{1-\gamma}$ for all $s \in S \setminus S^\mathrm{T}$ and $V_i^{\pi^\star}(s \mid \delta_i) = 0$ for all $s \in S^\mathrm{T}$. Now the original reward $R_i(s, a)$ is bounded in $[-h, h]$, which means an additional reward in this range in each step and, hence, an additional cumulative reward in the interval $\left[\frac{-h}{1-\gamma}, \frac{h}{1-\gamma}\right]$. Adding this to $\frac{H}{1-\gamma}$ gives the desired range $\left[\frac{H-h}{1-\gamma}, \frac{H+h}{1-\gamma}\right]$.

Hence, $V_i^{\pi^\star}(s \mid \delta_i) \in \left[\frac{H-h}{1-\gamma}, \frac{H+h}{1-\gamma}\right]$ for all $s \in S$. This further implies that, for any actions $a, b \in A$, it holds that

$$\sum_{s' \in S} P(s, a, s') \cdot V_i^{\pi^\star}(s' \mid \delta_i) \geq \sum_{s' \in S} P(s, b, s') \cdot V_i^{\pi^\star}(s' \mid \delta_i) - \frac{2h}{1-\gamma}. \tag{18}$$

---

[2]Full proofs and omitted proofs can all be found in the appendix.

We have
$$Q_i^{\pi^\star}(s, \pi^\star(s) \mid \delta_i) = R_i(s, \pi^\star(s)) + \delta_i(s, \pi^\star(s)) + \gamma \cdot \sum_{s' \in S} P(s, \pi^\star(s), s') \cdot V_i^{\pi^\star}(s' \mid \delta_i)$$

$$\geq -h + H + \gamma \cdot \sum_{s' \in S} P(s, \pi^\star(s), s') \cdot V_i^{\pi^\star}(s' \mid \delta_i)$$

$$\geq h + \epsilon + \gamma \cdot \sum_{s' \in S} P(s, a, s') \cdot V_i^{\pi^\star}(s' \mid \delta_i)$$

for any $a \in A$, where the last line follows by (18) and the fact that $H = \frac{2\gamma}{1-\gamma} \cdot h + 2h + \epsilon$. By definition, we have $\delta_i(s, a) = 0$ for all $a \neq \pi^\star(s)$. It follows that

$$Q_i^{\pi^\star}(s, \pi^\star(s) \mid \delta_i) \geq R_i(s, a) + \delta_i(s, a) + \gamma \cdot \sum_{s' \in S} P(s, a, s') \cdot V_i^{\pi^\star}(s' \mid \delta_i) + \epsilon$$

$$= Q_i^{\pi^\star}(s, a \mid \delta_i) + \epsilon.$$

Therefore, $\delta$ is a feasible scheme. $\qquad\square$

## B PoF Bounds

We analyze PoWEF first, and then PoEF and PoSEF.

### B.1 PoWEF

To analyze the PoWEF, we first derive its lower bound.

**Lemma B.1.** $\mathrm{PoWEF}(n, m, \lambda) = \Omega(\lambda \cdot \sqrt{m})$.

*Proof.* Consider the family of instances illustrated in Figure 3, and we consider the two-agent version of this example ($n = 2$) that consists of only agents 1 and 2. We show that the PoWEF of this particular family of instances is $\Omega(\lambda \cdot \sqrt{|S| \cdot |A|})$ to establish the lower bound of PoWEF.

First, the cost of teaching $\pi^\star$ without fairness constraints is at most 1. Indeed, without fairness constraints, $\pi^\star$ is already the optimal policy of agent 2 up to a robustness of $\epsilon$. As for agent 1, it suffices to set $\delta_1(c) = 1$. Hence, the total cost is 1.

Now consider the case with fairness constraints and suppose that $\delta = (\delta_1, \delta_2)$ is a WEF and feasible adjustment scheme. We argue that $\|\delta_1\| + \|\delta_2\| = \Omega(\lambda \cdot \sqrt{|S| \cdot |A|})$.

By symmetry, we can assume without loss of generality that each $\delta_i$ assigns the same reward for a state-action pair and its copies in the instance. Hence, in our analysis, it suffices to consider only the values associated with the original state-action pairs, which are representative of the values associated with their copies. Given this, we omit the state in the notation and write, e.g., $\delta_i(a) = \delta_i(s_l, a)$, as each action is associated with a unique state.

Consider the following two cases:

**Case 1:** $\delta_1(c) \leq 1/2$. Since $\delta_1$ incentivizes agent 1 to use the target policy $\pi^\star$, we have $Q_1^{\pi^\star}(s_r, d) \geq Q_1^{\pi^\star}(s_r, e) + \epsilon$, or equivalently,

$$\delta_1(d) + \epsilon + \frac{\gamma}{1-\gamma} \cdot (\delta_1(c) - 1) \geq \delta_1(e) + \epsilon.$$

Rearranging the terms gives

$$\delta_1(e) - \delta_1(d) \leq \frac{\gamma}{1-\gamma} \cdot (\delta_1(c) - 1) \leq -\frac{1}{2} \cdot \frac{\gamma}{1-\gamma}.$$

Note that for any two real numbers $x$ and $y$, we have $x^2 + y^2 \geq \frac{(x-y)^2}{2}$. Hence,

$$\|\delta_1\| \geq \sqrt{L} \cdot \sqrt{\delta_1^2(e) + \delta_1^2(d)} \geq \sqrt{L} \cdot \sqrt{\frac{(\delta_1(e) - \delta_1(d))^2}{2}}$$

$$\geq \sqrt{L} \cdot \frac{1}{\sqrt{8}} \cdot \frac{\gamma}{1-\gamma} = \Omega(\lambda \cdot \sqrt{|S| \cdot |A|}).$$

**Case 2:** $\delta_1(c) \geq 1/2$. By WEF, we have $\rho_1^{\pi^\star}(\delta_1) \geq \rho_1^{\pi^\star}(\delta_2)$ and $\rho_2^{\pi^\star}(\delta_2) \geq \rho_2^{\pi^\star}(\delta_1)$. Let $\varrho_i^{\pi^\star}(\delta_j) = \rho_i^{\pi^\star}(\delta_j) - \rho_i^{\pi^\star}(0)$, where $\rho_i^{\pi^\star}(0)$ denotes the agent's cumulative reward without any adjustment. Since now both agents 1 and 2 have the same discount factor $\gamma$, we have

$$\varrho_1^{\pi^\star}(\delta_j) = \varrho_2^{\pi^\star}(\delta_j)$$

for any $j$. Hence,

$$\rho_1^{\pi^\star}(\delta_1) \geq \rho_1^{\pi^\star}(\delta_2) \quad \Longrightarrow \quad \varrho_1^{\pi^\star}(\delta_1) \geq \varrho_1^{\pi^\star}(\delta_2) = \varrho_2^{\pi^\star}(\delta_2),$$

$$\text{and} \quad \rho_2^{\pi^\star}(\delta_2) \geq \rho_2^{\pi^\star}(\delta_1) \quad \Longrightarrow \quad \varrho_2^{\pi^\star}(\delta_2) \geq \varrho_2^{\pi^\star}(\delta_1) = \varrho_1^{\pi^\star}(\delta_1),$$

which means that $\varrho_1^{\pi^\star}(\delta_1) = \varrho_2^{\pi^\star}(\delta_2)$. Expanding this gives

$$\delta_1(a) + \left(\delta_1(d) + \frac{\gamma}{1-\gamma} \cdot \delta_1(c)\right) = \delta_2(a) + \left(\delta_2(d) + \frac{\gamma}{1-\gamma} \cdot \delta_2(c)\right). \tag{19}$$

Moreover, $\delta_2$ incentivizes agent 2 to use the target policy $\pi^\star$, so we have $Q_2^{\star\pi^\star}(s_l, a) \geq Q_2^{\pi^\star}(s_l, b) + \epsilon$, expanding which gives

$$\delta_2(a) + \epsilon \geq \delta_2(b) + \frac{\gamma}{1-\gamma} \cdot \delta_2(c) + \epsilon.$$

Combining (19) with the above equation gives

$$2 \cdot \delta_2(a) - \delta_2(b) + \delta_2(d) - \delta_1(a) - \delta_1(d) \geq \frac{\gamma}{1-\gamma} \cdot \delta_1(c) \geq \frac{1}{2} \cdot \frac{\gamma}{1-\gamma}.$$

Note that for any real numbers $x_1, \ldots, x_k$ and nonzero coefficients $a_1, \ldots, a_k$, we have $\sum_{i=1}^k x_i^2 \geq \left(\sum_{i=1}^k a_i \cdot x_i\right)^2 / \sum_{i=1}^k a_i^2$. It follows that

$$\|\delta_1\| + \|\delta_2\| \geq \sqrt{L} \cdot \sqrt{\delta_2^2(a) + \delta_2^2(b) + \delta_2^2(d) + \delta_1^2(a) + \delta_1^2(d)}$$

$$\geq \sqrt{L} \cdot \frac{1}{\sqrt{32}} \cdot \frac{\gamma}{1-\gamma}$$

$$= \Omega(\lambda \cdot \sqrt{|S| \cdot |A|}).$$

Therefore, in both cases, we have $\|\delta_1\| + \|\delta_2\| = \Omega(\lambda \cdot \sqrt{|S| \cdot |A|})$, which completes the proof. $\square$

**Lemma B.2.** $\mathrm{PoWEF}(n, m, \lambda) = O(\lambda \cdot \sqrt{m})$.

*Proof.* Suppose that without the fairness constraints the minimum costs for teaching $\pi^\star$ is $C_i$ for each agent $i \in [n]$; let $\widehat{\delta}_i$ be the adjustment achieving this minimum cost for each $i \in [n]$, and let $\widehat{\delta} = \left(\widehat{\delta}_i\right)_{i \in [n]}$. Hence, $\left|\widehat{\delta}_i(s, a)\right| \leq \left\|\widehat{\delta}_i\right\| = C_i$ for all $i$, $s$, and $a$.

We construct the following adjustment scheme $\delta = (\delta_i)_{i \in [n]}$ in an approach similar to that in the proof of Theorem 4.1. We let

$$\delta_i(s, a) = \begin{cases} 0, & \text{if } a = \pi^\star(s) \\ -\frac{2}{1-\gamma_i} \cdot C_i, & \text{otherwise} \end{cases} \tag{20}$$

for all $s \in S$ and $i \in [n]$. With this $\delta$, we have

$$\frac{\|\delta_i\|}{\left\|\widehat{\delta}_i\right\|} = \frac{\sqrt{\sum_{s \in S, a \in A}(\delta_i(s, a))^2}}{C_i} \leq \frac{\sqrt{|S| \cdot |A|} \cdot \frac{2}{1-\gamma_i} \cdot C_i}{C_i} = 2\lambda \cdot \sqrt{|S| \cdot |A|}. \tag{21}$$

Hence, the price of using $\delta$ is

$$\frac{\sum_{i \in [n]} \|\delta_i\|}{\sum_{i \in [n]} \left\|\widehat{\delta}_i\right\|} \leq 2\lambda \cdot \sqrt{|S| \cdot |A|} = O\left(\lambda \cdot \sqrt{|S| \cdot |A|}\right).$$

Therefore, it remains to argue that $\delta$ is feasible and WEF.

**Feasibility** Compare the differences in the V-values when $\widehat{\delta}$ and $\delta$ are applied. Since $V_i^{\pi^\star}$ only depends on the rewards of state-action pairs chosen by $\pi^\star$, we have

$$
\begin{aligned}
\left| V_i^{\pi^\star}(s \mid \delta_i) - V_i^{\pi^\star}\left(s \,\Big|\, \widehat{\delta}_i\right) \right| &= \left| \mathbb{E}\left[ \sum_{t=0}^{\infty} (\gamma_i)^t \cdot \left( \delta_i(s_t, \pi^\star(s_t)) - \widehat{\delta}_i(s_t, \pi^\star(s_t)) \right) \,\Bigg|\, s_0 \sim \mathbf{z}, \pi^\star \right] \right| \\
&= \left| \mathbb{E}\left[ \sum_{t=0}^{\infty} (\gamma_i)^t \cdot \widehat{\delta}_i(s_t, \pi^\star(s_t)) \,\Bigg|\, s_0 \sim \mathbf{z}, \pi^\star \right] \right| \\
&\leq \left| \sum_{t=0}^{\infty} (\gamma_i)^t \cdot C_i \right| \\
&= \frac{1}{1 - \gamma_i} \cdot C_i.
\end{aligned}
\tag{22}
$$

Now compare the Q-values. We have

$$
\begin{aligned}
Q_i^{\pi^\star}&(s, \pi^\star(s) \mid \delta_i) - Q_i^{\pi^\star}\left(s, \pi^\star(s) \,\Big|\, \widehat{\delta}_i\right) \\
&= \delta_i(s, \pi^\star(s)) - \widehat{\delta}_i(s, \pi^\star(s)) + \gamma_i \cdot \mathbb{E}_{x \sim P(s, \pi^\star(s), \cdot)}\left( V_i^{\pi^\star}(x \mid \delta_i) - V_i^{\pi^\star}\left(x \,\Big|\, \widehat{\delta}_i\right) \right) \\
&\geq -C_i - \frac{\gamma_i}{1 - \gamma_i} \cdot C_i \qquad\qquad\qquad\qquad\qquad\qquad\qquad\qquad \text{(by (22))} \\
&= -\frac{1}{1 - \gamma_i} \cdot C_i.
\end{aligned}
$$

Whereas for any $a \neq \pi^\star(s)$,

$$
\begin{aligned}
Q_i^{\pi^\star}(s, a \mid \delta_i) - Q_i^{\pi^\star}\left(s, a \,\Big|\, \widehat{\delta}_i\right) &= \delta_i(s, a) - \widehat{\delta}_i(s, a) + \gamma_i \cdot \mathbb{E}_{x \sim P(s, a, \cdot)}\left( V_i^{\pi^\star}(x \mid \delta_i) - V_i^{\pi^\star}\left(x \,\Big|\, \widehat{\delta}_i\right) \right) \\
&\leq \delta_i(s, a) - \widehat{\delta}_i(s, a) + \frac{\gamma_i}{1 - \gamma_i} \cdot C_i \qquad\qquad \text{(by (22))} \\
&\leq -\frac{2}{1 - \gamma_i} \cdot C_i + C_i + \frac{\gamma_i}{1 - \gamma_i} \cdot C_i \qquad \text{(by (20) and } \left\|\widehat{\delta}_i\right\| = C_i) \\
&= -\frac{1}{1 - \gamma_i} \cdot C_i
\end{aligned}
$$

Combining the above two equations gives

$$
Q_i^{\pi^\star}(s, \pi^\star(s) \mid \delta_i) - Q_i^{\pi^\star}(s, a \mid \delta_i) \geq Q_i^{\pi^\star}\left(s, \pi^\star(s) \,\Big|\, \widehat{\delta}_i\right) - Q_i^{\pi^\star}\left(s, a \,\Big|\, \widehat{\delta}_i\right)
$$

for any $s \in S$ and $a \neq \pi^\star(s)$. Indeed, since $\widehat{\delta}$ is feasible, by definition we have

$$
Q_i^{\pi^\star}\left(s, \pi^\star(s) \,\Big|\, \widehat{\delta}_i\right) \geq Q_i^{\pi^\star}\left(s, a \,\Big|\, \widehat{\delta}_i\right) + \epsilon
$$

if $a \neq \pi^\star(s)$. It then follows that

$$
Q_i^{\pi^\star}(s, \pi^\star(s) \mid \delta_i) - Q_i^{\pi^\star}(s, a \mid \delta_i) \geq \epsilon
$$

for all $a \neq \pi^\star(s)$. Since the choice of $i$ is arbitrary, by definition $\delta$ is feasible.

**Fairness** Indeed, since $\delta$ offers no additional reward for state-action pairs specified by the target policy $\pi^\star$, we have $\rho_i^{\pi^\star}(\delta_i) = \rho_i^{\pi^\star}(0) = \rho_i^{\pi^\star}(\delta_j)$ for all $i, j \in [n]$. Hence, $\delta$ is WEF. $\qquad\square$

### B.2  PoEF and PoSEF

Next we turn to PoEF and PoSEF.

**Lemma B.3.** $\mathrm{PoEF}(n, m, \lambda) = \Omega(\lambda \cdot n \cdot \sqrt{m})$.

*Proof.* We use the class of instances illustrated in Figure 3. Similarly to the two-agent version of the instances we used in the proof of Lemma B.1, the cost of teaching $\pi^\star$ without fairness constraints is at most 1. It suffices to set $\delta_1(s_*, c) = 1$ for agent 1, and keep the reward functions of all other agents as is since $\pi^\star$ is already optimal for agents $2, \ldots, n$ up to robustness $\epsilon$.

Now consider the case with fairness constraints. Suppose that $\delta = (\delta_1, \ldots, \delta_n)$ is an EF and feasible adjustment scheme, and without loss of generality $\delta_2 = \cdots = \delta_n$. We argue that $\sum_{i \in [n]} \|\delta_i\| = \Omega(\lambda \cdot n \cdot \sqrt{|S| \cdot |A|})$ to complete the proof.

Similarly to the argument in the proof of Lemma B.1, by symmetry we can assume without loss of generality that each $\delta_i$ assigns the same reward for a state-action pair and its copy, so we omit the state in the notation of $\delta_i$ and write, e.g., $\delta_i(a) = \delta_i(s_l, a)$, as each action is associated with a unique state that is not a copy.

Consider the following two cases:[3]

**Case 1:** $\delta_2(c) \geq 1/2$. Since $\delta_2$ incentivizes agent 2 to use the target policy $\pi^\star$, we have $Q_2^{\pi^\star}(s_l, a) \geq Q_2^{\pi^\star}(s_l, b) + \epsilon$, or equivalently,

$$\delta_2(a) + \epsilon \geq \delta_2(b) + \frac{\gamma}{1-\gamma} \cdot \delta_2(c) + \epsilon.$$

Rearranging the terms gives

$$\delta_2(a) - \delta_2(b) \geq \frac{\gamma}{1-\gamma} \cdot \delta_2(c) \geq \frac{1}{2} \cdot \frac{\gamma}{1-\gamma}.$$

For any real numbers $x$ and $y$, we have $x^2 + y^2 \geq \frac{(x-y)^2}{2}$. Hence,

$$\|\delta_2\| \geq \sqrt{L} \cdot \sqrt{\delta_2^2(a) + \delta_2^2(b)} \geq \sqrt{L} \cdot \sqrt{\frac{(\delta_2(a) - \delta_2(b))^2}{2}}$$

$$\geq \sqrt{L} \cdot \frac{1}{\sqrt{8}} \cdot \frac{\gamma}{1-\gamma} = \Omega(\lambda \cdot \sqrt{|S| \cdot |A|}).$$

**Case 2:** $\delta_2(c) \leq 1/2$. By EF, we have $\rho_1^{\pi^\star}(\delta_1) \geq \rho_1^{\pi^\star}(\delta_2)$ and $\rho_2^{\pi^\star}(\delta_2) \geq \rho_2^{\pi^\star}(\delta_1)$. The same as the proof of Lemma B.1, since the agents have the same discount factor, we have $\rho_1^{\pi^\star}(\delta_1) - \rho_1^{\pi^\star}(0) = \rho_2^{\pi^\star}(\delta_2) - \rho_1^{\pi^\star}(0)$, expanding which gives the following equation (the same as (19)).

$$\delta_1(a) + \left(\delta_1(d) + \frac{\gamma}{1-\gamma} \cdot \delta_1(c)\right) = \delta_2(a) + \left(\delta_2(d) + \frac{\gamma}{1-\gamma} \cdot \delta_2(c)\right). \tag{23}$$

Now by EF, agent 1 would not be better off if they were given $\delta_2$ and deviated to a policy $\pi$ with $\pi(s_l) = a$ and $\pi(s_r) = e$. Namely, $\rho_1^{\pi^\star}(\delta_1) \geq \rho_1^{\pi}(\delta_2)$, or equivalently

$$\delta_1(a) + \delta_1(d) + \frac{\gamma}{1-\gamma} \cdot (\delta_1(c) - 1) \geq \delta_2(a) + \delta_2(e).$$

Combining (23) with the above equation gives

$$\delta_2(d) - \delta_2(e) \geq \frac{\gamma}{1-\gamma} \cdot (1 - \delta_2(c)) \geq \frac{1}{2} \cdot \frac{\gamma}{1-\gamma}.$$

For any real numbers $x$ and $y$, we have $x^2 + y^2 \geq \frac{(x-y)^2}{2}$. It follows that

$$\|\delta_2\| \geq \sqrt{L} \cdot \sqrt{\delta_2^2(d) + \delta_2^2(e)}$$

$$\geq \sqrt{L} \cdot \frac{1}{\sqrt{8}} \cdot \frac{\gamma}{1-\gamma} = \Omega(\lambda \cdot \sqrt{|S| \cdot |A|}).$$

---

[3]The analysis of these two cases are similar to the analysis in the proof of Lemma B.1, but with a few differences. In particular, we focus on the adjustment for agent 2 in this proof and aim to show that $\|\delta_2\| = \Omega(\lambda \cdot \sqrt{|S| \cdot |A|})$ for both cases, whereas when WEF is considered we can only bound $\|\delta_1\|$ or $\|\delta_1\| + \|\delta_2\|$ in the proof of Lemma B.1.

Therefore, in both cases, we have $\|\delta_2\| = \Omega(\lambda \cdot \sqrt{|S| \cdot |A|})$. Since $\delta_2 = \delta_3 = \cdots = \delta_n$, we have

$$\text{cost}(\delta) \geq \sum_{i=2}^{n} \|\delta_i\| = \Omega(\lambda \cdot n \cdot \sqrt{|S| \cdot |A|}),$$

which completes the proof. $\qquad\square$

**Lemma B.4.** $\text{PoSEF}(n, m, \lambda) = O(\lambda \cdot n \cdot \sqrt{m})$.

*Proof.* The proof is similar to the proof of Lemma B.2. We penalize actions off the policy and let

$$\delta_i(s, a) = \begin{cases} 0, & \text{if } a = \pi^\star(s) \\ -\max_{j \in [n]} \frac{3}{1 - \gamma_j} \cdot C_j, & \text{otherwise} \end{cases}$$

for all $s \in S$ and $i \in [n]$. Hence, $\delta$ is SEF as all $\delta_i$'s are the same.

Similarly to (21), with this adjustment scheme $\delta$, we now have

$$\frac{\|\delta_i\|}{\max_{j \in [n]} \left\|\widehat{\delta}_j\right\|} = \frac{\sqrt{\sum_{s \in S, a \in A}(\delta_i(s, a))^2}}{\max_{j \in [n]} C_j} \leq 3\lambda \cdot \sqrt{|S| \cdot |A|}.$$

Hence, the price of using $\delta$ is

$$\frac{\sum_{i \in [n]} \|\delta_i\|}{\sum_{i \in [n]} \left\|\widehat{\delta}_i\right\|} \leq \frac{\sum_{i \in [n]} \|\delta_i\|}{\max_{i \in [n]} \left\|\widehat{\delta}_i\right\|} \leq n \cdot 3\lambda \cdot \sqrt{|S| \cdot |A|} = O\left(\lambda \cdot n \cdot \sqrt{m}\right).$$

The feasibility of $\delta$ follows by the same argument in the proof of Lemma B.2. $\qquad\square$

Summarizing the above lemmas, we get the following main theorem.

**Theorem 6.1.** $\text{PoWEF}(n, m, \lambda) = \Theta(\lambda \cdot \sqrt{m})$, $\text{PoEF}(n, m, \lambda) = \Theta(\lambda \cdot n \cdot \sqrt{m})$, *and* $\text{PoSEF}(n, m, \lambda) = \Theta(\lambda \cdot n \cdot \sqrt{m})$.

*Proof.* The bound of the PoWEF follows by the lower and upper bounds established in Lemmas B.1 and B.2.

Since SEF is a stronger requirement than EF, the bounds of the PoEF and PoSEF follow by Lemmas B.3 and B.4. $\qquad\square$

## C  PoF Bounds with Non-negativity

Since a feasible and fair solution may not exist with non-negative adjustments, we analyze the case where the agents have the same discount factor. The existence of a feasible fair solution is guaranteed in this case according to Theorem 4.3.

### C.1  PoWEF

**Lemma C.1.** $\text{PoWEF}(n, m, \lambda) = \Omega(\lambda \cdot n \cdot \sqrt{m})$ *when the scheme is required to be non-negative and all the agents have the same discount factor.*

*Proof.* Consider the family of instances illustrated in Figure 4. We show that the PoWEF of this particular family of instances is $\Omega(\lambda \cdot n \cdot \sqrt{m})$ to establish the lower bound.

First, the cost of teaching $\pi^\star$ without fairness constraints is at most 1: the target policy $\pi^\star$ is already optimal for agent 2, and it suffices to set $\delta_1(s_r, c) = 1$ to incentivize agent 1.

Now consider the case with fairness constraints and suppose that $\delta = (\delta_1, \ldots, \delta_n)$ is a WEF and feasible adjustment scheme. Without loss of generality, we can assume that $\delta_2 = \delta_3 = \cdots = \delta_n$, and we argue that $\|\delta_2\| = \Omega(\lambda \cdot \sqrt{m})$ to finish the proof.

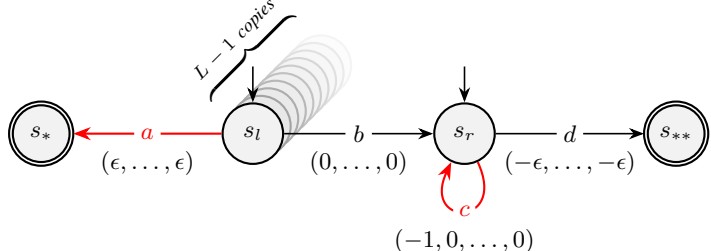

Figure 4: There are $n$ agents, all with discount factor $\gamma$. $A = \{a, b, c, d\}$ and all transitions are deterministic. The initial rewards are annotated on the corresponding edges, and they are identical for agents $2, \ldots, n$. There are $L - 1$ copies of $s_l$, each connected to $s_*$ and $s_r$ the same way $s_l$ is connected to these two states (and with the same initial rewards). The initial state distribution has probability $0.5/L$ on $s_l$ as well as each of its copies, and $0.5$ on $s_r$. The target policy is highlighted in red: $\pi^\star(s) = a$ for $s = s_l$ and its copies, and $\pi^\star(s_r) = c$.

By symmetry, we can assume without loss of generality that each $\delta_i$ assigns the same reward for a state-action pair and its copies in the instance. Hence, it suffices to consider only the values associated with the original state-action pairs, and we omit the state in the notation and write, e.g., $\delta_i(a) = \delta_i(s_l, a)$, as each action is associated with a unique state.

Consider the following two cases.

**Case 1:** $\delta_2(c) \geq 1/2$. Since $\delta_2$ incentivizes agent 2 to use the target policy $\pi^\star$, we have $Q_2^{\pi^\star}(s_l, a) \geq Q_2^{\pi^\star}(s_l, b) + \epsilon$, or equivalently,

$$\delta_2(a) + \epsilon \geq \delta_2(b) + \frac{\gamma}{1 - \gamma} \cdot \delta_2(c) + \epsilon.$$

Since $\delta_2$ is non-negative and by assumption $\delta_2(c) \geq 1/2$ in this case, we get that $\delta_2(a) \geq \frac{1}{2} \cdot \frac{\gamma}{1-\gamma}$. By symmetry this also holds for all copies of action $a$. It follows that

$$\|\delta_2\| \geq \frac{\sqrt{L}}{2} \cdot \frac{\gamma}{1 - \gamma} = \Omega(\lambda\sqrt{m}).$$

**Case 2:** $\delta_2(c) \leq 1/2$. Note that since $\delta_1$ is non-negative and it incentivizes agent 1 to select action $c$, it must be that $\delta_1(c) \geq 1$. By WEF, we have $\rho_2^{\pi^\star}(\delta_2) \geq \rho_2^{\pi^\star}(\delta_1)$, which means

$$0.5 \cdot (\epsilon + \delta_2(a)) + 0.5 \cdot \frac{1}{1 - \gamma} \cdot \delta_2(c) \geq 0.5 \cdot (\epsilon + \delta_1(a)) + 0.5 \cdot \frac{1}{1 - \gamma} \cdot \delta_1(c).$$

Rearranging the terms and using the facts that $\delta_1(c) \geq 1$ and all adjustments are non-negative, we get that $\delta(a) \geq \frac{1}{2} \cdot \frac{1}{1-\gamma}$ and

$$\|\delta_2\| \geq \frac{\sqrt{L}}{2} \cdot \frac{1}{1 - \gamma} = \Omega(\lambda\sqrt{m}).$$

Therefore, in both cases, $\|\delta_2\| = \Omega(\lambda \cdot \sqrt{m})$. Since $\delta_2 = \delta_3 = \cdots = \delta_n$, we have $\text{cost}(\delta) \geq \sum_{i=2}^n \|\delta_i\| = \Omega(\lambda \cdot n \cdot \sqrt{m})$, which completes the proof. □

**Lemma C.2.** $\text{PoWEF}(n, m, \lambda) = O(\lambda \cdot n \cdot \sqrt{m})$ *when the scheme is required to be non-negative and all the agents have the same discount factor.*

*Proof.* Suppose that without the fairness constraints the minimum costs for teaching $\pi^\star$ is $C_i$ for each agent $i \in [n]$; let $\widehat{\delta}_i$ be the adjustment achieving this minimum cost for each $i \in [n]$, and let $\widehat{\delta} = \left(\widehat{\delta}_i\right)_{i \in [n]}$. Hence, $\left|\widehat{\delta}_i(s, x)\right| \leq \left\|\widehat{\delta}_i\right\| = C_i$ for all $i$, $s$, and $x$.

Note that since the agents have the same discount factor, the improvement $\varrho^{\pi^\star}$ of the cumulative reward is the same for all $i \in [n]$:

$$\varrho^{\pi^\star}\left(\widehat{\delta}_j\right) := \rho_i^{\pi^\star}\left(\widehat{\delta}_j\right) - \rho_i^{\pi^\star}(0).$$

For each $i \in [n]$, we let

$$H_i = (1-\gamma) \cdot \left(\max_{j \in [n]} \varrho^{\pi^\star}\left(\widehat{\delta}_j\right) - \varrho^{\pi^\star}\left(\widehat{\delta}_i\right)\right).$$

Then we construct the following adjustment scheme $\delta = (\delta_i)_{i \in [n]}$:

$$\delta_i(s,a) = \begin{cases} \widehat{\delta}_i(s,a) + H_i + \frac{\gamma}{1-\gamma} \cdot H_i \cdot \sum_{s' \in S^{\mathrm{T}}} P(s,a,s'), & \text{if } a = \pi^\star(s) \\ 0, & \text{otherwise} \end{cases} \tag{24}$$

For any $s$ and $a$, we have

$$\delta_i(s,a) \le \widehat{\delta}_i(s,a) + \frac{1}{1-\gamma} \cdot H_i$$

$$\le \widehat{\delta}_i(s,a) + \max_{j \in [n]} \varrho^{\pi^\star}\left(\widehat{\delta}_j\right) \le \frac{2}{1-\gamma} \cdot \max_{j \in [n]} C_j,$$

where we use $\widehat{\delta}_i(s,a) \le \max_{j \in [n]} C_j$ and $\varrho^{\pi^\star}\left(\widehat{\delta}_j\right) \le \frac{1}{1-\gamma} \cdot C_j$, and the latter is due to the fact that the agent gets an additional reward of at most $C_j$ at each time step when $\widehat{\delta}_j$ is applied. It follows that the price of using $\delta$ is

$$\frac{\mathrm{cost}(\delta)}{\mathrm{cost}\left(\widehat{\delta}\right)} \le \frac{\sum_{i \in [n]} \|\delta_i\|}{\max_{i \in [n]} \left\|\widehat{\delta}_i\right\|} \le \frac{n \cdot 2\lambda \cdot \max_{i \in [n]} C_i \cdot \sqrt{|S| \cdot |A|}}{\max_{i \in [n]} C_i} = O\left(\lambda \cdot n \cdot \sqrt{m}\right).$$

Therefore, it remains to argue that $\delta$ is feasible and WEF.

Now that non-negativity is imposed, we can assume without loss of generality that $\widehat{\delta}_i(s,a) = 0$ for all $s \in S$ and $a \neq \pi^\star(s)$. Therefore, the way $\delta$ is defined in (24) is equivalent to adding an additional reward $H_i$ to agent $i$ on top of what is already offered by $\widehat{\delta}_i$. The term $\frac{\gamma}{1-\gamma} \cdot H_i \cdot \sum_{s' \in S^{\mathrm{T}}} P(s,a,s')$ adjusts the reward in consideration of subsequent terminal states, so that it is as if the process continues forever with an additional $H_i$ offered at every subsequent step. Consequently, this improves the V-value of every non-terminal state by $\frac{1}{1-\gamma} \cdot H_i$, i.e., for every $s \in S \setminus S^{\mathrm{T}}$ and every pair $i, j \in [n]$ we have

$$V_i^{\pi^\star}(s \mid \delta_j) = V_i^{\pi^\star}\left(s \mid \widehat{\delta}_j\right) + \frac{1}{1-\gamma} \cdot H_i. \tag{25}$$

**Feasibility**  Since the V-values of all non-terminal states increase by the same amount, $\delta$ remains feasible. Specifically, since $\widehat{\delta}$ is feasible, we have

$$Q_i^{\pi^\star}\left(s, \pi^\star(s) \mid \widehat{\delta}_i\right) \ge Q_i^{\pi^\star}\left(s, a \mid \widehat{\delta}_i\right) + \epsilon$$

for all $s$ and $a \neq \pi^\star(s)$. Now compare $\delta$ and $\widehat{\delta}$. We have

$$Q_i^{\pi^\star}\left(s, \pi^\star(s) \mid \delta_i\right) - Q_i^{\pi^\star}\left(s, \pi^\star(s) \mid \widehat{\delta}_i\right)$$

$$= \delta_i(s, \pi^\star(s)) - \widehat{\delta}_i(s, \pi^\star(s)) + \gamma \cdot \mathbb{E}_{x \sim P(s, \pi^\star(s), \cdot)}\left(V_i^{\pi^\star}(x \mid \delta_i) - V_i^{\pi^\star}\left(x \mid \widehat{\delta}_i\right)\right)$$

$$= \delta_i(s, \pi^\star(s)) - \widehat{\delta}_i(s, \pi^\star(s)) + \gamma \cdot \sum_{x \in S \setminus S^{\mathrm{T}}} P(s, \pi^\star(s), x) \cdot \left(V_i^{\pi^\star}(x \mid \delta_i) - V_i^{\pi^\star}\left(x \mid \widehat{\delta}_i\right)\right)$$

$$+ \gamma \cdot \sum_{x \in S^{\mathrm{T}}} P(s, \pi^\star(s), x) \cdot \left(V_i^{\pi^\star}(x \mid \delta_i) - V_i^{\pi^\star}\left(x \mid \widehat{\delta}_i\right)\right)$$

$$= H_i + \gamma \cdot \sum_{x \in S \setminus S^{\mathrm{T}}} P(s, \pi^\star(s), x) \cdot \left(V_i^{\pi^\star}(x \mid \delta_i) - V_i^{\pi^\star}\left(x \mid \widehat{\delta}_i\right)\right)$$

$$+ \gamma \cdot \sum_{x \in S^{\mathrm{T}}} P(s, \pi^\star(s), x) \cdot \left(\frac{1}{1-\gamma} \cdot H_i + V_i^{\pi^\star}(x \mid \delta_i) - V_i^{\pi^\star}\left(x \mid \widehat{\delta}_i\right)\right),$$

Using (25) and the fact that the V-values of all the terminal states are zero, we further get that

$$Q_i^{\pi^\star}\left(s, \pi^\star(s) \mid \delta_i\right) - Q_i^{\pi^\star}\left(s, \pi^\star(s) \,\middle|\, \widehat{\delta}_i\right)$$

$$= H_i + \gamma \cdot \sum_{x \in S \setminus S^{\mathrm{T}}} P(s, \pi^\star(s), x) \cdot \frac{1}{1-\gamma} \cdot H_i + \gamma \cdot \sum_{x \in S^{\mathrm{T}}} P(s, \pi^\star(s), x) \cdot \frac{1}{1-\gamma} \cdot H_i$$

$$= \frac{1}{1-\gamma} \cdot H_i.$$

Next, consider actions $a \neq \pi^\star(s)$. We have

$$Q_i^{\pi^\star}\left(s, a \mid \delta_i\right) - Q_i^{\pi^\star}\left(s, a \,\middle|\, \widehat{\delta}_i\right) = \delta_i(s, a) - \widehat{\delta}_i(s, a) + \gamma \cdot \mathbb{E}_{x \sim P(s, a, \cdot)}\left(V_i^{\pi^\star}\left(x \mid \delta_i\right) - V_i^{\pi^\star}\left(x \,\middle|\, \widehat{\delta}_i\right)\right)$$

$$\leq \gamma \cdot \mathbb{E}_{x \sim P(s, a, \cdot)}\left(V_i^{\pi^\star}\left(x \mid \delta_i\right) - V_i^{\pi^\star}\left(x \,\middle|\, \widehat{\delta}_i\right)\right)$$

$$\leq \frac{\gamma}{1-\gamma} \cdot H_i.$$

It follows that

$$Q_i^{\pi^\star}\left(s, \pi^\star(s) \mid \delta_i\right) - Q_i^{\pi^\star}\left(s, a \mid \delta_i\right) \geq Q_i^{\pi^\star}\left(s, \pi^\star(s) \,\middle|\, \widehat{\delta}_i\right) - Q_i^{\pi^\star}\left(s, a \,\middle|\, \widehat{\delta}_i\right) \geq \epsilon$$

for any $s \in S$ and $a \neq \pi^\star(s)$. Since the choice of $i$ is arbitrary, $\delta$ is feasible.

**Fairness** By definition $\rho_i^{\pi^\star}(\delta_j) = V_i^{\pi^\star}(\mathbf{z} \mid \delta_j)$, where $\mathbf{z}$ is the initial state distribution. Using (25), we then get that

$$\rho_i^{\pi^\star}(\delta_j) = \rho_i^{\pi^\star}\left(\widehat{\delta}_j\right) + \frac{1}{1-\gamma} \cdot H_i$$

$$= \rho_i^{\pi^\star}\left(\widehat{\delta}_j\right) + \max_{i' \in [n]} \varrho^{\pi^\star}\left(\widehat{\delta}_{i'}\right) - \varrho^{\pi^\star}\left(\widehat{\delta}_i\right)$$

$$\leq \rho_i^{\pi^\star}\left(\widehat{\delta}_i\right) + \max_{i' \in [n]} \varrho^{\pi^\star}\left(\widehat{\delta}_{i'}\right) - \varrho^{\pi^\star}\left(\widehat{\delta}_i\right) \qquad \text{(as } \widehat{\delta} \text{ is WEF)}$$

$$= \rho_i^{\pi^\star}(0) + \max_{i' \in [n]} \varrho^{\pi^\star}\left(\widehat{\delta}_{i'}\right)$$

for all $i, j \in [n]$. The right side does not depend on $j$, which means $\rho_i^{\pi^\star}(\delta_i) = \rho_i^{\pi^\star}(\delta_j)$, for all $j$, so $\delta$ is WEF. $\qquad \square$

## C.2 PoEF and PoSEF

**Lemma C.3.** $\mathrm{PoEF}(n, m, \lambda) = \Omega(\lambda^2 \cdot n \cdot \sqrt{m})$ *when the scheme is required to be non-negative and all the agents have the same discount factor.*

*Proof.* Consider the family of instances illustrated in Figure 5. We show that the PoEF of this particular family of instances is $\Omega(\lambda^2 \cdot n \cdot \sqrt{m})$ to establish the lower bound.

First, the cost of teaching $\pi^\star$ without fairness constraints is at most 2: the target policy $\pi^\star$ is already optimal for agents $3, \ldots, n$, and it suffices to set $\delta_1(s_l, c) = 1$ to incentivize agent 1.

Now consider the case with fairness constraints and suppose that $\delta = (\delta_1, \ldots, \delta_n)$ is EF and feasible. Without loss of generality, we can assume that $\delta_3 = \cdots = \delta_n$, and we argue that $\|\delta_2\| = \Omega(\lambda^2 \cdot n \cdot \sqrt{m})$ to finish the proof.

By symmetry, we can assume without loss of generality that each $\delta_i$ assigns the same reward for a state-action pair and its copies in the instance. Hence, it suffices to consider only the values associated with the original state-action pairs, and we omit the state in the notation and write, e.g., $\delta_i(a) = \delta_i(s_l, a)$, as each action is associated with a unique state.

Observe that the structure of the MDP is symmetric with respect to agents 1 and 2. Hence, without loss of generality, we can also assume the same symmetry in $\delta$:

$$\delta_1(a) = \delta_2(h), \quad \delta_1(h) = \delta_2(a), \quad \delta_1(c) = \delta_2(f), \quad \text{and } \delta_1(f) = \delta_2(c). \tag{26}$$

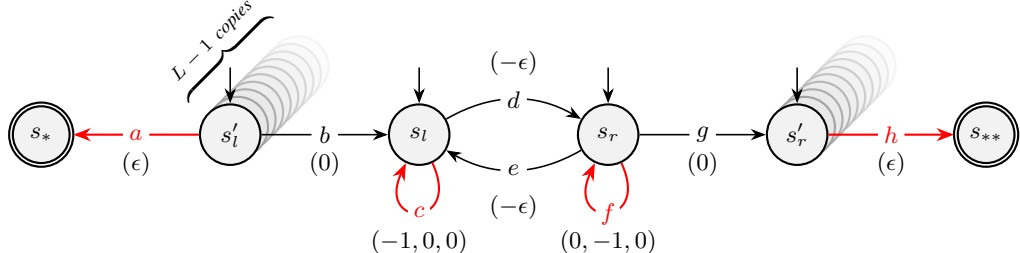

Figure 5: There are $n$ agents, all with discount factor $\gamma$. $A = \{a, b, c, \ldots, h\}$ and all transitions are deterministic. The initial rewards of agents 1, 2, and 3 are annotated on the corresponding edges (if there is only one number, then all the agents have the same reward). Agents $4, \ldots, n$ have the same reward function as agent 3. There are $L - 1$ copies of $s'_l$ and $s'_r$, each connected to the other states the same way $s_l$ and $s_r$ are connected (and with the same initial rewards). The initial state distribution has probability $0.25/L$ on each of $s'_l$ and $s'_r$ as well as each of their copies, and $0.25$ on each of $s_l$ and $s_r$. The target policy is highlighted in red: $\pi^\star(s'_l) = a$, $\pi^\star(s_l) = c$, $\pi^\star(s_r) = f$, and $\pi^\star(s'_r) = h$ (and the same for the corresponding copies).

Next, we first show that $\delta_1(c) \geq \frac{1}{1-\gamma} - \epsilon$ and $\delta_1(f) \geq \frac{1}{1-\gamma} - \epsilon$. Since $\delta$ incentivizes agent 1 to take action $c$ instead of $d$, we have $Q_1^{\pi^\star}(s_l, c \mid \delta_1) \geq Q_1^{\pi^\star}(s_l, d \mid \delta_1) + \epsilon$, expanding which gives

$$\frac{1}{1-\gamma} \cdot (\delta_1(c) - 1) \geq -\epsilon + \frac{\gamma}{1-\gamma} \cdot \delta_1(f) + \epsilon,$$

or

$$\delta_1(c) \geq \gamma \cdot \delta_1(f) + 1. \tag{27}$$

Since $\delta$ is EF, agent 1 cannot be better off with the following policy $\pi$ and $\delta_2$: $\pi(s_l) = d$ and $\pi(s) = \pi^\star(s)$ for all other $s$. Namely, $\rho_1^\pi(\delta_2) \leq \rho_1^{\pi^\star}(\delta_1)$, or

$$(\delta_2(a) + \epsilon) \overbrace{-\epsilon + \frac{\gamma}{1-\gamma} \cdot \delta_2(f)}^{V_1^\pi(s_l \mid \delta_2)} + \frac{1}{1-\gamma} \cdot \delta_2(f) + (\delta_2(h) + \epsilon)$$

$$\leq (\delta_1(a) + \epsilon) + \frac{1}{1-\gamma} \cdot (\delta_1(c) - 1) + \frac{1}{1-\gamma} \cdot \delta_1(f) + (\delta_1(h) + \epsilon),$$

where we omit the initial probability $0.25$ as the coefficients on both sides of the equation. Applying (26), we can reduce the above equation to

$$1 + \gamma \cdot \delta_1(c) - (1 - \gamma) \cdot \epsilon \leq \delta_1(f).$$

Combining (27) with the above equation gives

$$\delta_1(f) \geq \gamma^2 \cdot \delta_1(f) + \gamma + 1 - (1 - \gamma) \cdot \epsilon,$$

$$\implies \delta_1(f) \geq \frac{1}{1-\gamma} - \frac{\epsilon}{1+\gamma} \geq \frac{1}{1-\gamma} - \epsilon;$$

$$\text{and} \quad \delta_1(c) \geq \gamma \cdot \delta_1(f) + 1 \geq \frac{1}{1-\gamma} - \epsilon.$$

The remainder of the proof is then similar to the proof of Lemma C.1 (where we had $\delta_1(c) \geq 1$ but now $\delta_1(c) \geq \frac{1}{1-\gamma} - \epsilon$). We analyze the following three cases.

**Case 1:** $\delta_3(c) \geq \lambda/2$. Since $\delta_3$ incentivizes agent 3 to use the target policy $\pi^\star$, we have $Q_3^{\pi^\star}(s'_l, a) \geq Q_3^{\pi^\star}(s'_l, b) + \epsilon$, or equivalently,

$$\delta_3(a) + \epsilon \geq \delta_3(b) + \frac{\gamma}{1-\gamma} \cdot \delta_3(c) + \epsilon.$$

Since $\delta_3$ is non-negative and by assumption $\delta_3(c) \geq \lambda/2$ in this case, we get that $\delta_3(a) \geq \frac{\lambda}{2} \cdot \frac{\gamma}{1-\gamma}$. By symmetry this also holds for all copies of action $a$. It follows that

$$\|\delta_3\| \geq \sqrt{L} \cdot \frac{\lambda}{2} \cdot \frac{\gamma}{1-\gamma} = \Omega(\lambda^2 \sqrt{m}).$$

**Case 2: $\delta_3(f) \geq \lambda/2$.** Applying the same arguments for Case 1 gives $\|\delta_3\| = \Omega(\lambda^2 \sqrt{m})$ in this case.

**Case 3: $\delta_3(c) \leq \lambda/2$ and $\delta_3(f) \leq \lambda/2$.** We have shown that $\delta_1(c) \geq \frac{1}{1-\gamma} - \epsilon$ and $\delta_1(f) \geq \frac{1}{1-\gamma} - \epsilon$. By WEF, we have $\rho_3^{\pi^\star}(\delta_3) \geq \rho_3^{\pi^\star}(\delta_1)$, which means

$$(\delta_3(a) + \epsilon) \;+\; \frac{1}{1-\gamma} \cdot \delta_3(c) \;+\; \frac{1}{1-\gamma} \cdot \delta_3(f) \;+\; (\delta_3(h) + \epsilon)$$

$$\geq (\delta_1(a) + \epsilon) \;+\; \frac{1}{1-\gamma} \cdot \delta_1(c) \;+\; \frac{1}{1-\gamma} \cdot \delta_1(f) \;+\; (\delta_1(h) + \epsilon).$$

Rearranging the terms and using non-negativity and the facts that $\delta_1(c) \geq \frac{1}{1-\gamma} - \epsilon$ and $\delta_1(f) \geq \frac{1}{1-\gamma} - \epsilon$, as well as the assumption that $\delta_3(c) \leq \lambda/2$ and $\delta_3(f) \leq \lambda/2$ in this case, we get that

$$\delta_3(a) + \delta_3(h) \geq \left(\frac{1}{1-\gamma}\right)^2 - \frac{2\epsilon}{1-\gamma} = \lambda^2 - 2\epsilon \cdot \lambda.$$

It follows that

$$\|\delta_3\| \geq \sqrt{\frac{L \cdot (\delta_3(a) + \delta_3(h))^2}{2}} = \Omega(\lambda^2 \sqrt{m}).$$

Therefore, in all cases, $\|\delta_3\| = \Omega(\lambda^2 \cdot \sqrt{m})$. Since $\delta_3 = \cdots = \delta_n$, we have $\text{cost}(\delta) \geq \sum_{i=3}^{n} \|\delta_i\| = \Omega(\lambda^2 \cdot n \cdot \sqrt{m})$, which completes the proof. $\qquad\square$

**Lemma C.4.** $\text{PoSEF}(n, m, \lambda) = O(\lambda^2 \cdot n \cdot \sqrt{m})$ *when the scheme is required to be non-negative and all the agents have the same discount factor.*

*Proof.* Let $\gamma_1 = \cdots = \gamma_n = \gamma$. Suppose that without the fairness constraints, the minimum costs for teaching $\pi^\star$ is $C_i$ for each agent $i \in [n]$; let $\widehat{\delta}_i$ be the adjustment achieving this minimum cost for each $i \in [n]$, and let $\widehat{\delta} = \left(\widehat{\delta}_i\right)_{i \in [n]}$. Since the schemes are non-negative, we have $0 \leq \widehat{\delta}_i(s, a) \leq C_i$ for all $i$, $s$, and $a$.

Now consider SEF and the following adjustment scheme (similar to (17)), where we let $H = \frac{1}{1-\gamma} \max_{i \in [n]} C_i$ and $S^{\mathrm{T}}$ be the set of terminal states.

$$\delta_i(s, a) = \begin{cases} H + \frac{\gamma}{1-\gamma} \cdot H \cdot \sum_{s' \in S^{\mathrm{T}}} P(s, a, s'), & \text{if } a = \pi^\star(s) \\ 0, & \text{otherwise} \end{cases} \tag{28}$$

As defined above, $\delta$ is non-negative, and $\delta_i$ is identical for all $i \in [n]$, so $\delta$ is SEF. Moreover, we have $0 \leq \delta_i(s, a) \leq \frac{1}{1-\gamma} \cdot H$ for all $i$, $s$, and $a$. Hence,

$$\frac{\text{cost}(\delta)}{\text{cost}\left(\widehat{\delta}\right)} \leq \frac{\sum_{i \in [n]} \|\delta_i\|}{\max_{i \in [n]} \left\|\widehat{\delta}_i\right\|} \leq \frac{n \cdot \lambda \cdot H \cdot \sqrt{|S| \cdot |A|}}{\max_{i \in [n]} C_i} = O\left(\lambda^2 \cdot n \cdot \sqrt{m}\right).$$

It remains to argue that $\delta$ is also feasible.

Consider an arbitrary agent $i$. We first argue that

$$V_i^{\pi^\star}(s \,|\, \delta_i) = V_i^{\pi^\star}(s \,|\, 0) + \frac{1}{1-\gamma} \cdot H \tag{29}$$

for all $s \in S \setminus S^{\mathrm{T}}$, where $V_i^{\pi^\star}(s \,|\, 0)$ denotes the original value function when no adjustment is provided. Indeed, since the V-function is additive for two reward functions, it suffices to argue that

in a process where the $\delta_i$ is the reward function, the corresponding V-values are $\frac{1}{1-\gamma} \cdot H$ for every $s \in S \setminus S^{\mathrm{T}}$. This can be verified via the Bellman equation: The V-values are 0 for all the terminal states, whereas for the non-terminal states, the term $\frac{\gamma}{1-\gamma} \cdot H \cdot \sum_{s' \in S^{\mathrm{T}}} P(s, a, s')$ makes it as if the process continues forever with a reward $H$ generated in every subsequent step, whereby the V-values are exactly $\frac{1}{1-\gamma} \cdot H$. Hence, (29) then follows.

Next consider $\widehat{\delta}$, we have

$$V_i^{\pi^\star}\left(s \,\middle|\, \widehat{\delta}_i\right) = V_i^{\pi^\star}(s) + \mathbb{E}\left[\sum_{t=0}^{\infty} (\gamma_i)^t \cdot \widehat{\delta}_i(s_t, \pi^\star(s_t)) \,\middle|\, s_0 \sim \mathbf{z}, \pi^\star\right].$$

Hence,

$$V_i^{\pi^\star}(s \mid 0) \leq V_i^{\pi^\star}\left(s \,\middle|\, \widehat{\delta}_i\right) \leq V_i^{\pi^\star}(s \mid 0) + \frac{1}{1-\gamma} \cdot C, \tag{30}$$

where we let $C = \max_{i \in [n]} C_i$. The first inequality follows by the non-negativity of $\widehat{\delta}$, and the second follows by the fact that $\widehat{\delta}_i(s, a) \leq C_i \leq C$ for all $i$, $s$, and $a$.

Compare the differences in the Q-values when $\widehat{\delta}$ and $\delta$ are applied. We have

$$Q_i^{\pi^\star}\left(s, \pi^\star(s) \mid \delta_i\right) - Q_i^{\pi^\star}\left(s, \pi^\star(s) \,\middle|\, \widehat{\delta}_i\right)$$

$$= \delta_i(s, \pi^\star(s)) - \widehat{\delta}_i(s, \pi^\star(s)) + \gamma \cdot \mathbb{E}_{x \sim P(s, \pi^\star(s), \cdot)}\left(V_i^{\pi^\star}(x \mid \delta_i) - V_i^{\pi^\star}\left(x \,\middle|\, \widehat{\delta}_i\right)\right)$$

$$= \delta_i(s, \pi^\star(s)) - \widehat{\delta}_i(s, \pi^\star(s)) + \gamma \cdot \sum_{x \in S \setminus S^{\mathrm{T}}} P(s, \pi^\star(s), x) \cdot \left(V_i^{\pi^\star}(x \mid \delta_i) - V_i^{\pi^\star}\left(x \,\middle|\, \widehat{\delta}_i\right)\right)$$

$$+ \gamma \cdot \sum_{x \in S^{\mathrm{T}}} P(s, \pi^\star(s), x) \cdot \left(V_i^{\pi^\star}(x \mid \delta_i) - V_i^{\pi^\star}\left(x \,\middle|\, \widehat{\delta}_i\right)\right)$$

$$= H - \widehat{\delta}_i(s, \pi^\star(s)) + \gamma \cdot \sum_{x \in S \setminus S^{\mathrm{T}}} P(s, \pi^\star(s), x) \cdot \left(V_i^{\pi^\star}(x \mid \delta_i) - V_i^{\pi^\star}\left(x \,\middle|\, \widehat{\delta}_i\right)\right)$$

$$+ \gamma \cdot \sum_{x \in S^{\mathrm{T}}} P(s, \pi^\star(s), x) \cdot \left(\frac{1}{1-\gamma} \cdot H + V_i^{\pi^\star}(x \mid \delta_i) - V_i^{\pi^\star}\left(x \,\middle|\, \widehat{\delta}_i\right)\right),$$

where the last equality follows by replacing $\delta_i(s, \pi^\star(s))$ according to (28). Note that for all terminal states $x \in S^{\mathrm{T}}$, we have $V_i^{\pi^\star}(x \mid \delta_i) = V_i^{\pi^\star}\left(x \,\middle|\, \widehat{\delta}_i\right) = 0$. Moreover, using (29) and (30), we have $V_i^{\pi^\star}(x \mid \delta_i) - V_i^{\pi^\star}\left(x \,\middle|\, \widehat{\delta}_i\right) \geq \frac{1}{1-\gamma} \cdot (H - C)$. Hence, the above equation continues as:

$$Q_i^{\pi^\star}\left(s, \pi^\star(s) \mid \delta_i\right) - Q_i^{\pi^\star}\left(s, \pi^\star(s) \,\middle|\, \widehat{\delta}_i\right)$$

$$\geq H - \widehat{\delta}_i(s, \pi^\star(s)) + \gamma \sum_{x \in S \setminus S^{\mathrm{T}}} P(s, \pi^\star(s), x) \cdot \frac{1}{1-\gamma} \cdot (H - C) + \gamma \sum_{x \in S^{\mathrm{T}}} P(s, \pi^\star(s), x) \cdot \frac{1}{1-\gamma} \cdot H$$

$$\geq H - C + \frac{\gamma \cdot H}{1-\gamma} - \frac{\gamma \cdot C}{1-\gamma}$$

$$\geq \frac{\gamma}{1-\gamma} \cdot H.$$

Next, we consider actions $a \neq \pi^\star(s)$.

$$Q_i^{\pi^\star}\left(s, a \mid \delta_i\right) - Q_i^{\pi^\star}\left(s, a \,\middle|\, \widehat{\delta}_i\right) = \delta_i(s, a) - \widehat{\delta}_i(s, a) + \gamma \cdot \mathbb{E}_{x \sim P(s, a, \cdot)}\left(V_i^{\pi^\star}(x \mid \delta_i) - V_i^{\pi^\star}\left(x \,\middle|\, \widehat{\delta}_i\right)\right)$$

$$\leq \gamma \cdot \mathbb{E}_{x \sim P(s, a, \cdot)}\left(V_i^{\pi^\star}(x \mid \delta_i) - V_i^{\pi^\star}\left(x \,\middle|\, \widehat{\delta}_i\right)\right)$$

$$\leq \frac{\gamma}{1-\gamma} \cdot H,$$

where the last transition follows by (28) and (30).

Combining the above two equations gives

$$Q_i^{\pi^\star}\left(s, \pi^\star(s) \mid \delta_i\right) - Q_i^{\pi^\star}\left(s, a \mid \delta_i\right) \geq Q_i^{\pi^\star}\left(s, \pi^\star(s) \,\middle|\, \widehat{\delta}_i\right) - Q_i^{\pi^\star}\left(s, a \,\middle|\, \widehat{\delta}_i\right)$$

for any $s \in S$ and $a \neq \pi^\star(s)$. Indeed, since $\widehat{\delta}$ is feasible, by definition we have

$$Q_i^{\pi^\star}\left(s, \pi^\star(s) \,\middle|\, \widehat{\delta}_i\right) \geq Q_i^{\pi^\star}\left(s, a \,\middle|\, \widehat{\delta}_i\right) + \epsilon.$$

It then follows that

$$Q_i^{\pi^\star}\left(s, \pi^\star(s) \mid \delta_i\right) - Q_i^{\pi^\star}\left(s, a \mid \delta_i\right) \geq \epsilon$$

for all $a \neq \pi^\star(s)$. Since the choice of $i$ is arbitrary, $\delta$ is feasible. $\qquad\square$

Summarizing the above two lemmas, we get the following result.

**Theorem 6.2.** *When the scheme is required to be non-negative and all the agents have the same discount factor, it holds that* $\mathrm{PoWEF}(n, m, \lambda) = \Theta(\lambda \cdot n \cdot \sqrt{m})$, $\mathrm{PoEF}(n, m, \lambda) = \Theta(\lambda^2 \cdot n \cdot \sqrt{m})$, *and* $\mathrm{PoSEF}(n, m, \lambda) = \Theta(\lambda^2 \cdot n \cdot \sqrt{m})$.

*Proof.* Lemmas C.1 and C.2 establish the bound of the PoWEF.

Since SEF is a stronger requirement than EF, Lemmas C.3 and C.4 establish the bounds of the PoEF and PoSEF. $\qquad\square$