# OpenReview forum: "Envy-free Policy Teaching to Multiple Agents"
_NeurIPS.cc/2022/Conference — NeurIPS 2022 Accept_

### Official Review · Reviewer_sVfc · 2022-07-03

**Rating:** 7
**Confidence:** 4
**Soundness:** 4 excellent
**Presentation:** 4 excellent
**Contribution:** 3 good

**Summary:**

The authors here consider a model of envy-free policy teaching.  At a high level, some number of agents are independently maximizing their own personal reward function over a MDP with the same state-space and transition dynamics.  The principal wants to teach a specified target policy to each individual, and does so by modifying the agent's individual reward functions with "incentives" (either additive penalties or costs for taking specific actions in certain states).  Naively, these incentives can lead to envy behaviour (where one agent is given a large increase cost for taking a certain action which other agents could be envious of).  The authors introduce a concept of envy freeness, and show that an EF solution need not exist when penalties are not allowed, but exist otherwise.  They also show an explicit worst-case bound on the price of fairness which scales linearly with the discount factor, size of the MDP, and number of individuals.

To be more concrete, the authors consider a setting with $n$ agents.  Each agent is faced with an MDP $(S,A,R_i, P, \gamma_i)$ where $S,A$ are the set of states and actions, $R_i$ is individual $i$'s reward function, $P$ the shared transition dynamics, and $\gamma_i$ the per individual discount factor.  The principal wants each agent $i$ to execute a pre-specified target policy $\pi^\star$ (which need not be the optimal policy).  This can be done by adding an additional "reward adjustment function" $\delta_i : S \times A \rightarrow R$ which is provided to agent $i$ upon taking action $a$ in state $s$.  The goal of the principal then is to find a cost-efficient way of teaching this target policy to all of the individuals, potentially by minimizing the expected cumulative payment across all of the individuals.

Intuitively, these incentive schemes need not be fair.  The authors then go on to model three definitions of envy-freeness (denoted EF, SEF, and CF).  Focusing on the first one, envy-freeness, stipulates that agent $i$ receives more reward under their adjustment scheme $\delta_i$ under policy $\pi^\star$ than under the adjustment scheme $\delta_j$.  This is a natural fairness yardstick, essentially ensuring that each agent prefers their own adjustment scheme to the adjustment scheme of any other agent.  With this the authors show two main results:
1. First the authors show that there exists a fair feasible solution, and the solution is efficiently computable by a linear program.  However, under the additional restriction that $\delta \geq 0$ the authors show that an adjustment need not exist, unless additionally the discount factors are the same.
2. Next the authors show explicit bounds on the price of fairness, essentially the increased cost in designing incentives which are additionally envy free.
These results are then repeated (and adjusted) under the other two stronger envy-freeness guarantees.

**Questions:**

### Questions
1. The LP based approach for finding the reward adjustment function only works in the setting with known reward and transition dynamics.  Have the authors thought about extending the techniques to a learning framework, where the MDP primitives and the reward adjustments need to be learned jointly?
2. The Price of Fairness guarantees presented are worst-case, do the authors have any intuition on more "instance-specific" losses, potentially also observed with the different choice of cost measures as well?

### Minor Comments
- Related work section was great!
- Line 143 does not discuss the way to aggregate the cost measures across agents (although this is introduced later)
- Line 163 $j$ for $\beta$
- Found the OPT notation in Equation 4 a bit strange

**Limitations:**

The authors do not address the potential societal impact of their work explicitly in the paper. However, while practically motivated, the model considered here by the authors is mostly theoretical. Any concrete implementation of envy-free incentive design in a real-world setting has additional challenges around scalability, choice of fairness metrics (with potential adverse side-effects for other metrics), adversarial behavior, etc, which is not addressed in this theoretical work.

**Strengths And Weaknesses:**

### Strengths
- Model + Theoretical Results: The paper makes strong modeling and methodological contributions by introducing the envy-free set-up for incentive design for multi-agent MDPs.  This extends the well-studied envy-freeness concepts in fair resource allocation to policy design in MDPs.
- Quality of Results: The authors provide detailed theoretical results both outlining 1) when EF incentives can be computed, 2) how to compute them, 3) at what loss, providing a complete story behind the envy-free incentive design model.
- Relation to Existing Literature: The authors do a good job relating the current analysis and model to the RL, reward shaping, and fair resource allocation literature.

### Weaknesses
- Weakness of Results: One of the larger weaknesses in the results is the strong assumptions for computationally finding the envy-free incentives.  The LP based approach requires exact knowledge of the reward and transitions, which is impractical for many settings.
- Writing + Practical Motivation: The authors should include a practical motivating example to help instantiate the discussion and help the model become more clear.

---

> ### Author Response · Authors · 2022-08-02
> **Responses to Reviewer sVfc**
>
> We thank the reviewer for the insightful comments. Our responses are as follows.
>
> - **Motivation Example.** One concrete motivating example of our work is a classroom teaching setting, where a teacher teaches a particular skill (represented as a target policy in an MDP) to multiple students. The teacher might adopt personalized teaching schemes for the students and reward (or penalize) them differently for taking the same actions. It is usually important that students feel they are treated equally by the teacher in such scenarios. Similar examples include a principal-agent setting, such as the interaction between a company who outsources a task described as a policy in an MDP to multiple contractors. The company wants to incentivize the contractors to follow a desired policy, while fairness might also be an important consideration as a beneficial factor for long-term partnerships. We have added these examples in the revision.
>
> - **Learning Setting.** We appreciate the reviewer’s suggestion of the learning setting of our problem and find it an interesting and natural direction to work on. We did not consider the learning setting in the current work but have the following preliminary thoughts. The computation methods we presented in Section 5 resemble formulation for policy teaching in single-agent settings, and amount to solving MDPs with additional linear constraints on the policy space. Hence, we expect standard techniques for solving similar learning problems to be applicable based on the formulation we presented.
>
> - **Instance-specific PoF.** The price of fairness of a particular instance can be computed directly: by computing the costs for teaching with and without fairness constraints, and then calculate the ratio.
>
> We are more than happy to take any other questions from the reviewer.

---

> > ### Comment · Reviewer_sVfc · 2022-08-05
> > **Response**
> >
> > Thanks for the comments - agree that incorporating the learning setting would be an interesting direction for future work.  Potentially interesting as well is how the true envy constraints (defined in terms of reward under true MDP of a particular reward scheme) get carried over to the approximate / learned MDP!
> >
> > One last comment, especially since there are no numerical simulations in the main section, it might make sense to adjust the second paragraph in the introduction to give an example more on a data-driven level instead of a philosophical example with teaching schemes in a classroom.

---

> > > ### Author Response · Authors · 2022-08-09
> > > **Thank you for your feedback**
> > >
> > > Thank you for your further feedback and useful suggestions!
> > >
> > > We do not have an example based on real data so far but do have the following more concrete classroom teaching instance in mind.
> > >
> > > Take language teaching as an example. It can be modeled as an MDP where a state represents a student's overall skill and is encoded as the student's performance on different components such as listening, reading, speaking, and writing. Students may come with different interests over the components: e.g., some are more interested in reading, some enjoy speaking, and some are just not a fan of any of them. The interests define their innate reward functions. Actions of the students represent how much effort they invest on each component, and it is desired that students always put more effort on components that they are currently weaker at. The target policy is defined accordingly. The teacher can assign additional credits to incentivize the students to follow the target policy (e.g., credits that can be used to exchange snacks, or that will be considered in termly evaluations). Similar interactions may also happen with other types of training programs in various domains (e.g., in sports training). And they can happen in physical classrooms, or virtual classrooms such as language learning apps (e.g., in Duolingo, a credit system is used where credits can be used to unlock next learning levels).
> > >
> > > We will revise the paper to include these details.

---

### Official Review · Reviewer_UkNf · 2022-07-11

**Rating:** 6
**Confidence:** 3
**Soundness:** 3 good
**Presentation:** 4 excellent
**Contribution:** 3 good

**Summary:**

The paper studies how a teacher could teach a target policy to multiple agents, subject to fairness constraints. In particular, the paper focuses on envy-free fairness, under which all agents would prefer their own adjustments over anyone else's.

The contributions of the paper are theoretical in nature and include:
1. Proving the existence of CF solutions when adjustments could be negative.
    1.a While also showing that when adjustments are limited to nonnegative ones, there exist settings under which no EF solution exists.
    1.b But when agents have the same discount factor, there also exists CF solutions with nonnegative adjustments.
2. Showing how optimal fair solutions may be attained with suitable constraints.
3. Deriving the price of fairness under different settings.

**Questions:**

My concerns and questions are mostly on the significance and motivation side.
1. Are there practical use cases for the problem setting and algorithms discussed in this paper? Are there real-world scenarios that could be easily translated to policy teaching with fairness constraint? As the paper features a novel combination of fairness and policy teaching, it would be greatly appreciated if it could detail the motivation behind such combination.


**Limitations:**

The authors have adequately addressed the limitations and potential negative societal impact.

**Strengths And Weaknesses:**

**Strengths**:
1. Originality:
    - The problem setting proposed by the authors is novel, combining both policy teaching and fairness. Related works are adequately cited and given credit to and the difference between this work and prior works is made clear.
2. Quality:
    - I have checked the supplement and could not identify any glaring mistake in the proofs.
    - Moreover, the paper is keenly aware of the potential negative social impact of the work from the policy poisoning perspective.
3. Clarity:
    - Overall the paper is well-written, but there are some formatting inconsistencies (expanded later).

**Weaknesses**:
1. Clarity:
    - Definitions 3.4 and 3.5 are not capitalized, whereas Definitions 3.1 - 3.3 are.
2. Significance:
    - As the paper is a novel combination of two areas, fairness and policy teaching, the significance of the work is unclear to me. Solving the constrained optimization programs outlined in Section 5 requires that the teachers know the transition probabilities as well as all agents' reward functions. It is unclear to me what kind of problems fall into this problem setting.
        - For instance, in the motivating example of ride-share platforms encouraging good behavior from passengers, it is not clear to me what would a corresponding MDP look like.

---

> ### Author Response · Authors · 2022-08-02
> **Responses to Reviewer UkNf**
>
> We thank the reviewer for the insightful comments. Our responses are as follows.
>
> - **Style Issue.** We will capitalize the terms in the definitions to make the styles consistent.
>
> - **Motivation Example.** One concrete motivating example of our work is a classroom teaching setting, where a teacher teaches a particular skill (represented as a target policy in an MDP) to multiple students. The teacher might adopt personalized teaching schemes for the students and reward (or penalize) them differently for taking the same actions. It is usually important that students feel they are treated equally by the teacher in such scenarios. Similar examples include a principal-agent setting, such as the interaction between a company who outsources a task described as a policy in an MDP to multiple contractors. The company wants to incentivize the contractors to follow a desired policy, while fairness might also be an important consideration as a beneficial factor for long-term partnerships. We have added these examples in the revision.
>
> We are more than happy to take any other questions from the reviewer.

---

> > ### Comment · Reviewer_UkNf · 2022-08-05
> > **Score Updated**
> >
> > Thank the authors for the response. After reading the motivating example provided by the authors and the other reviewers' comments, I  decided to update the score.

---

> > > ### Author Response · Authors · 2022-08-05
> > > **Re: Score Updated**
> > >
> > > Thank you for reading our response and updating the score! Please feel free to let us know if you have any other questions.

---

### Official Review · Reviewer_LYiW · 2022-07-12

**Rating:** 7
**Confidence:** 3
**Soundness:** 4 excellent
**Presentation:** 4 excellent
**Contribution:** 3 good

**Summary:**

The authors present a theoretical contribution, connecting concepts from the theory of fair division to the problem of policy teaching in reinforcement learning. More precisely, they introduce several constraints inspired by "envy-freeness" to the setting of policy teaching by reward adjustment for heterogeneous reinforcement learning agents, each of which is interacting independently in the same single-agent Markov decision process. The authors prove that envy-free adjustment schemes exist in general if and only if negative reward adjustment are available. They go on to show that the problem of finding such schemes is tractable in principle, being a convex optimization problem with linear constraints. Finally, they prove asymptotically tight bounds for the cost of fairness, analogous to the price of anarchy from game theory.

**Questions:**

1. What are the clearest motivations for this work? How do you see it impacting the field in the future?

2. Is the convex optimization problem in Section 5 soluble for "large" MDPs (e.g. gridworlds or extensive-form games)?

3. Why is the cumulative measure of cost a good one to use? It feels like there is an underlying "implementation" you have in mind for the teacher being an RL agent, but this isn't explicitly stated. If that is the case, how might your results be impacted by the teacher and "students" being co-optimized?

**Limitations:**

The authors could be a bit more precise about how these theoretical results might translate into empirical outcomes, sketching out any limitations along the way. Having said this, they do a good job of expressing the problem in terms of convex optimization, and of proving asymptotically tight complexity bounds.

The authors comment on a potential risk from reward poisoning, but point out that bad actors may not be interested in fairness. It is not quite clear how true this claim is: state-sponsored reward poisoning may indeed have its risks assessed for fairness, for instance. It would be great to see a stronger argument for whether methods of implementing safety against reward poisoning are likely compatible with, or antithetical to, fairness.

**Strengths And Weaknesses:**

Strengths

- The paper is well written, with a clear abstract, good notation, well articulated definitions, and rigorous proofs.
- Theorem 4.2 is particularly interesting and significant, with implications for how one might think about reward shaping choices in both single-agent and multi-agent settings.
- The example figures are well-motivated and well-explained. Figure 3 gives good intuition for the price of fairness scaling laws.
- To the best of my knowledge, the work is a novel intersection of these two important areas of research (RL and fair division).

Weaknesses

- The motivation for the paper could be more strongly articulated. Are the authors imagining that this line of research leads to better mechanism design for fair outcomes among populations of humans? Or are there particular settings where there will be heterogeneous artificial agents and fairness of reward shaping will be an important consideration?
- In Figure 1, it is not clear why s_l is optimal on the right hand side. Should it not have a small positive reward \epsilon to make it a unique rewarding terminal state.
- The related work is a little thin on the ground. I would love to see a stronger connection drawn with inverse RL, which is strongly related to policy teaching. Some Inverse RL papers are cited, but I think a sentence or two here eliciting the distinction from / similarity to inverse RL in their setting would be worthwhile.
- The conclusion does not sketch particularly clear lines for future work. In particular, might the authors be interested in extending these results to the interactive multi-agent setting? It is known that different reward shaping is required to solve different multi-agent problems (see e.g. https://arxiv.org/pdf/1803.08884.pdf) and indeed different reward schemes have lately been used in various mechanism design settings (see e.g. https://arxiv.org/abs/2004.13332, https://www.nature.com/articles/s41562-022-01383-x).
- Some minor errors in grammar e.g. "policy" line 123, "\delta_j" line 163, "limits" line 287 etc. I recommend an additional proof read.
- There are no empirical results. In particular, it is not quite clear how feasible it is to solve the convex optimization problem in Section 5 for an MDP of real-world relevance.

---

> ### Author Response · Authors · 2022-08-02
> **Responses to Reviewer LYiW**
>
> We thank the reviewer for the insightful comments. Our responses are as follows.
>
> **Re: Main Questions**
>
> - **Motivation Example.** One concrete motivating example of our work is a classroom teaching setting, where a teacher teaches a particular skill (represented as a target policy in an MDP) to multiple students. The teacher might adopt personalized teaching schemes for the students and reward (or penalize) them differently for taking the same actions. It is usually important that students feel they are treated equally by the teacher in such scenarios. Similar examples include a principal-agent setting, such as the interaction between a company who outsources a task described as a policy in an MDP to multiple contractors. The company wants to incentivize the contractors to follow a desired policy, while fairness might also be an important consideration as a beneficial factor for long-term partnerships. We have added these examples in the revision.
>
> - **Impact to the field.** We think our work introduces a framework for studying fairness and the concept of envy-freeness to research on policy teaching. We believe that it could inspire future work aimed at a wider range of settings or specific applications along this line.
>
> - **Convex Optimization.** The convex optimization applies to gridworlds or extensive form games as long as the state and action spaces are finite (i.e., tabular setting). Theoretically, such convex optimizations can be efficiently approximated within a given error bound. In practice, there are also existing optimization solvers that are readily applicable. The exact practical scalability is not a focus of our current work, though we agree it is an interesting question to ask.
>
> - **Cost Measure.** The cumulative reward is a natural measure that is also widely considered in the literature of policy teaching (or reward poisoning), and more generally, in planning and sequential decision-making tasks. If the teacher and the agents are co-optimized and their total cumulative reward is considered, we can replace the teacher’s reward function with this social welfare, so the same results regarding the computation, existence of EF schemes, and price of EFness will follow.
>
> **Re: Other Comments**
>
> - **Figure 1.** For simplicity we omitted the robustness guarantee epsilon in this example (as this notion is introduced later in Section 2). We will add a remark in the example.
>
> - **Related literature.** We thank the reviewer for pointing out the related literature on inverse RL and interactive multi-agent learning. We will follow the suggestion and add one or two sentences to emphasize the connections. The connection to interactive multi-agent problems is very interesting. We do not have any concrete ideas so far about possible extensions along this line but take interest in exploring it.
>
> - **Limitations on fairness and reward poisoning.** We think that a malicious party who conducts reward poisoning usually doesn't care about whether poisoning is done fairly among the agents. We are not so sure about the specific objective of state-sponsored reward poisoning the reviewer mentioned. If the objective is to protect agents against reward poisoning while ensuring that protection is provided fairly among the agents, then it seems to be a defense-attack problem from a defender’s perspective and hence somewhat different than our current model (though very related). We are happy to answer any follow-up questions of the reviewer about this point (and any other questions as well).

---

> > ### Comment · Reviewer_LYiW · 2022-08-08
> > **Acknowledgement of Author Rebuttal**
> >
> > Having read the author rebuttal and the other reviews, I am happy to keep my score at 7. This feels to me like an interesting and well-written paper. The main weakness remains the motivation, but there are sufficiently many connections to other fields here for fruitful discussions around future work to take place at the conference.
> >
> > Finally, I would like to echo another reviewer in advising the authors to seek a more precise motivation for their work. They have stated that they would like to see this research find application in a classroom setting, but how exactly? Are they envisaging the development of a teacher agent that would teach humans? Or are they seeking to model the teaching interaction so as to better recommend to teachers the way to disseminate information? Under these circumstances, what are the main limitations of the policy teaching framework, and what problems remain to be solved?

---

> > > ### Author Response · Authors · 2022-08-09
> > > **Thank you for your feedback**
> > >
> > > Thank you for your further feedback and useful suggestions!
> > >
> > > One concrete example, as we also replied to Reviewer sVfc, is as follows.
> > >
> > > Take language teaching as an example. It can be modeled as an MDP where a state represents a student's overall skill and is encoded as the student's performance on different components such as listening, reading, speaking, and writing. Students may come with different interests over the components: e.g., some are more interested in reading, some enjoy speaking, and some are just not a fan of any of them. The interests define their innate reward functions. Actions of the students represent how much effort they invest on each component, and it is desired that students always put more effort on components that they are currently weaker at. The target policy is defined accordingly. The teacher can assign additional credits to incentivize the students to follow the target policy (e.g., credits that can be used to exchange snacks, or that will be considered in termly evaluations). Similar interactions may also happen with other types of training programs in various domains (e.g., in sports training). And they can happen in physical classrooms, or virtual classrooms such as language learning apps (e.g., in Duolingo, a credit system is used where credits can be used to unlock next learning levels). We will revise the paper to include these details.
> > >
> > > The limitations of our current work with respect to the above task include primarily the assumption of full information and the setup of a unique target policy, which were also pointed out in the discussions with the other reviewers. Moreover, target policies that specify a set of permissible actions for each state may also be useful for our model to capture a richer set of tasks. The current setup only allows one action for each state.
> > >
> > > Our current considerations are more about reward design, and we have not considered teaching by way of strategic information dissemination. But indeed this is also a relevant domain, where fairness might be an interesting subject to study.
> > >
> > > We thank the reviewer again for asking many insightful questions which have helped improve the work greatly.

---

### Official Review · Reviewer_CDiZ · 2022-07-12

**Rating:** 4
**Confidence:** 4
**Soundness:** 3 good
**Presentation:** 3 good
**Contribution:** 2 fair

**Summary:**

The paper looks at the problem of policy teaching, where a teacher wants to incentivise agents to follow a particular policy, under the constraint of incentives being fair in the sense that they are envy-free. The paper first proposes some definitions of envy-free (EF) teaching and fairness in this context. They then prove a negative and a positive result characterizing when EF policies exist. They discuss the tractability of finding cost-minimizing solutions under two natural cost functions, and introduce the concept of a price of fairness - the additional cost of the teaching incurred by following the EF conditions


**Questions:**

The result that EF strategies may exist for non-negative incentives, if agents’ discount factors are not equal, receives a large amount of the total text. I think theorem 4.3 and section 6 are more important contributions and could be prioritized.


**Limitations:**

The authors have a good argument as to why the connection to reward poisoning isn’t a concern, which is actually repeated (I’m not sure that’s necessary).

The authors don’t discuss / reference the limitations of EF as a concept. While it is a common concept in game theory, it may not correspond to what we consider to be fair or equitable. E.g. if two people are both given the same large incentive to use their right hand to complete a task, in the sense of the paper this is envy free (indeed, ‘completely fair’), but a left handed person would probably not agree that it was actually fair.

It seems a little unlikely that the EF concept cannot be extended to a situation where agents are incentivized to follow different strategies, because in general effective EF incentives may not exist. This limits the potential of future work unfortunately.


**Strengths And Weaknesses:**

The paper was written clearly, the contributions were well spelt out, and the preliminaries detailed thoroughly. (A minor comment on presentation: I found the notations x_i = \delta_i(s_y, a) (and vica-versa) a little confusing, it seemed like a more natural choice was x_i  = \delta_i(s_x, a)).

The results given establish some useful properties about the new concepts introduced, and I feel like they provide some intuition about the nature of the EF teaching policies. I liked that the authors considered the case of only positive reward interventions.

The limitation that the agents are operating in independent environments is an unfortunate but clearly necessary limitation. I think there are practical examples where this is a sensible model though, so I would suggest adding some examples to this effect to motivate the setting.

I would have liked to see some analysis of the case where agents are being taught to follow different policies. There are some reasonably obvious negative results in this case (if agents A and B prefer different strategies, you can’t swap their strategies around without one of them envying the other), but are there any cases where (s)EF conditions can be met?

The definition of EF is quite weak: it amounts more or less to all agents receiving the same reward adjustments when following the incentivised policy. But agents would adapt, so I think EF is much better captured by the SEF definition. I would recommend naming EF - Weak EF and Strong EF to EF to reflect this.

I found section 5 less interesting, it more or less only points out that the constraints are linear ones.

---

> ### Author Response · Authors · 2022-08-02
> **Responses to Reviewer CDiZ**
>
> We thank the reviewer for the insightful comments. Our responses are as follows.
>
> **Re: Choice of Problem Setting**
>
> - We appreciate that the reviewer suggested several other related settings where the same fair policy teaching problem could also be asked and studied. We agree that these are interesting settings but would also like to point out that **it might not be feasible to study all these settings in one conference paper.** In particular, **to derive our results already requires substantial non-trivial arguments (with an 8-page-long appendix)**. We selected the setting which we think is the most basic (yet technically non-trivial) and representative to introduce the concept of envy-freeness to research on policy teaching.
>
> **Re: Motivation Example**
>
> - One particular example where the agents operate in separate environments while having the same target policy is the classroom teaching setting, where a teacher teaches a particular skill (represented as a target policy in an MDP) to multiple students. Similarly in a principal-agent setting, a company who outsources a task to multiple contractors needs to ensure that the contracts are made fairly. We have added these examples in the revision.
>
> - In the setting where the agents operate in the same environment, if the target policy is still the same for all the agents, then our results apply as well. In the setting where different agents might have different target policies (whether they are in the same environment or not), to design an appropriate EF notion may require additional application-specific knowledge as the agents’ situations may not be directly comparable. (Imagine for example to decide whether an HR and an engineer in a company are rewarded fairly.)  Since our work is the first to introduce envy-freeness to policy teaching problems, we think it might be more appropriate to base our study on a model that is free from such domain-specific details.
>
> **Re: Limitation of the EF concept**
>
> - Our requirement that an adjustment scheme is feasible partially addresses this limitation. Take the example provided by the reviewer: with the requirement of feasibility, a reward needs to be provided to a left handed agent in the first place to incentivize them to use their right hand. Though we agree that this outcome may not be completely fair when examined under other notions of fairness/equality/equity that might be more appropriate for this scenario. We will note this limitation of the EF concept in the paper.
>
> **Re: Other comments on naming and structuring**
>
> - We thank the reviewer for the suggestions on the naming of the EF notions and organization of the contents. We follow the suggestions and renamed the EF notions in the revision. We also shortened some proofs in the main paper to proof sketches and added details about Theorem 4.3 and Section 6.
>
> We are more than happy to take any other questions from the reviewer.

---

### Meta-Review · Area_Chair_dK12 · 2022-08-20

**Recommendation:** Accept
**Confidence:** Less certain

**Metareview:**

I agree with a reviewer that one of the existence results is a bit trivial: it
simply says that if you can change payoffs arbitrarily then sure, you can cause
anything to happen because you can just penalize everything outside the policy
by infinity. The same can be said for Theorem 4.3. As that same reviewer points
out, the results in Section 5 are simply pointing out linearity.

From a technical perspective, the results that are non-trivial are: Theorem 4.2,
and the price of fairness bounds. From a technical perspective I would say the
results are a bit thin though.

Moreover, the applications are also fairly weak: the authors failed to argue
convincingly either in the rebuttal or the paper that there are really credible
applications of this.

Thus, the argument for acceptance rests primarily on the following: the model is
interesting, and it is a fairly clean and easy to understand problem. It is
possible that this paper would spur interesting follow-up work.


**Award:**

No

---

### Decision · Program_Chairs · 2022-09-14

Accept